# ProRefiner: an entropy-based refining strategy for inverse protein folding with global graph attention

Xinyi Zhou [1], Guangyong Chen [2] ✉, Junjie Ye[3], Ercheng Wang[2,4], Jun Zhang [5], Cong Mao[5], Zhanwei Li[2], Jianye Hao[3], Xingxu Huang[2], Jin Tang[2] & Pheng Ann Heng[1,2]

Inverse Protein Folding (IPF) is an important task of protein design, which aims to design sequences compatible with a given backbone structure. Despite the prosperous development of algorithms for this task, existing methods tend to rely on noisy predicted residues located in the local neighborhood when generating sequences. To address this limitation, we propose an entropy-based residue selection method to remove noise in the input residue context. Additionally, we introduce ProRefiner, a memory-efficient global graph attention model to fully utilize the denoised context. Our proposed method achieves state-of-the-art performance on multiple sequence design benchmarks in different design settings. Furthermore, we demonstrate the applicability of ProRefiner in redesigning Transposon-associated transposase B, where six out of the 20 variants we propose exhibit improved gene editing activity.

Computational Protein Design, which is to design proteins with specific structures or functions[1], has been a powerful tool to prompt the exploration of sequence or topology space not yet visited by evolutionary process[2–4] and discover proteins with better properties[5]. It has enabled success in membrane protein design[6], enzyme design[7], etc. As one of the sub-tasks of Computational Protein Design, Inverse Protein Folding (IPF), the problem of finding amino acid sequences that can fold into a given three-dimensional (3D) structure[8], is of great importance as hosting a particular function often presupposes acquiring a specific backbone structure.

How to model and utilize residue interactions has been the focus of various IPF algorithms. In traditional methods, energy functions are designed to approximate backbone-sequence compatibility. Residue-pair interaction modeling is usually derived from databases by leveraging statistical preferences for particular residue pairs in a simplified local environment to estimate inter-residue energies[5,9,10]. The increasing computational complexity limits the statistical estimation of multi-residue interactions that are conditional on a more fine-grained representation of the local environment[10,11].

In recent years, deep learning has been widely and successfully applied to protein structure modeling and prediction[12,13], due to its ability to automatically learn complex non-linear many-body interactions from data. There have been efforts to solve IPF with deep learning[4,14,15]. Early methods often model protein structures as sequences of independent residues[16,17] or atom point clouds[4,15] and adopt a non-autoregressive decoding scheme as demonstrated in Fig. 1a. Their independence assumption prevents them from learning complex residue interactions and limits their performance. Some recent works use proximity graphs to represent protein structures, where residues are nodes and residue interactions are directly modeled as edges. Typically, a masked encoder-decoder architecture with an autoregressive decoding method is used (shown in Fig. 1b)[18–21]. Recently, a similar decoding scheme has been proposed in ABACUS-R (shown in Fig. 1c)[22]. This method assumes all neighbor residue types are known when decoding a central residue. Starting from a random initial sequence, it updates residues recursively based on their neighborhood until convergence. However, the dependency on previous predictions has proven to be prone to the error accumulation

[1]Department of Computer Science and Engineering, The Chinese University of Hong Kong, Central Ave, Hong Kong, China. [2]Zhejiang Lab, Kechuang Avenue, Hangzhou, China. [3]Noah's Ark Lab, Huawei, Shenzhen, China. [4]College of Pharmaceutical Sciences, Zhejiang University, Hangzhou, China. [5]State Key Laboratory of Reproductive Medicine, Nanjing Medical University, Nanjing, China. ✉e-mail: gychen@zhejianglab.com

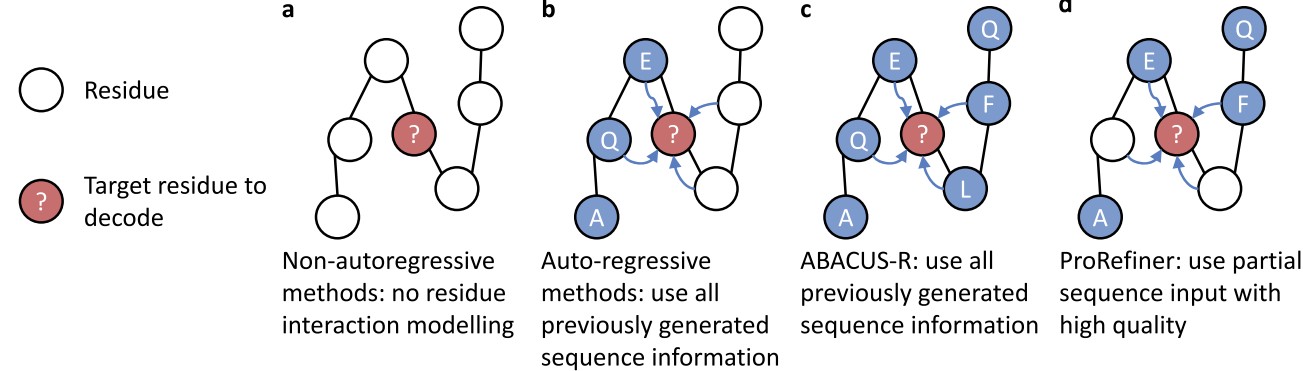

**Fig. 1 | Different ways of utilizing interresidue features. a** Nonautoregressive decoding scheme. **b** Autoregressive decoding scheme. **c** Decoding scheme proposed in ABACUS-R. **d** Our proposed method.

problem[23,24]. Noisy residue information is introduced into the context and propagated through the graph structure, while recovering a target residue would be easier and more accurate if more high-quality residue interactions were available and utilized.

We summarize the above issues as the selection and utilization of high-quality residue interactions. To address these issues, we propose a protein sequence design model ProRefiner. The model is tasked with BERT[25]-like sequence inpainting conditioned on protein structures. Specifically, we mask random residues on sequences during training. The model takes the masked partial sequences and backbone structures as input and learns to reconstruct the whole sequence. During inference, the input partial sequence to ProRefiner could be constructed in two ways. In partial sequence design scenarios where only some residues need to be designed, the remaining residues can naturally serve as an oracle partial input sequence, whereas in entire sequence design settings, we introduce an entropy-based residue selection technique to utilize predictions from existing models. Specifically, given a sequence generated by an inverse folding model, which is referred to as the base model, we include residues with highly confident predictions into the partial sequence and mask less valuable predictions with low confidence. Here we use entropy to approximate the base model's confidence in its predictions. In our experiments, the precision among the residues with the lowest 10% entropy is around 99%. By masking out high-entropy residues, a significant amount of noise can be effectively removed from the input residue environment. ProRefiner then generates the whole sequence in one shot based on the denoised partial sequence. Compared to previous left-to-right sequence design models, ProRefiner learns to exploit global residue interactions by training with partial sequence input. Its one-shot generation manner, together with the proposed residue selection technique, ensures higher-quality residue interactions and faster generation speed. It can be used as an add-on module to refine the results of existing methods.

ProRefiner's model architecture is a stack of memory-efficient global graph attention layers as shown in Fig. 2. Attention mechanism has been proven effective in modeling global dependencies for sequential data[26]. However, adapting attention to the graph domain is challenging. Specifically, the attention mechanism calculates attention weights between any two nodes based on their features. For graphs, this requires storing and manipulating a square matrix of size equal to the number of nodes, which neglects the sparsity of graph structures and increases the memory complexity to quadratic in terms of node count, posing scalability issues[27,28]. Some methods circumvent this by confining attention within node neighborhoods, losing the global view that makes attention powerful[18,29,30].

Moreover, these methods do not fully utilize edge features, as they only contribute to attention computation without the ability to be updated or influence node feature updates[18,27,29,31]. However, edge features have been proven to be critical in protein structure modeling[20]. In summary, to address these limitations, we aim to design an attention-based model tailored for graphs that (1) is memory efficient, (2) maintains a global view of dependencies, and (3) fully incorporates edge features.

In particular, a *K*-nearest neighbor graph is constructed from the backbone structure. Informative node features and edge features are extracted (detailed in Section Methods), and send to a stack of memory-efficient global graph attention layers. In each layer, every residue node globally attends to other residues. An attention score is calculated from both node and edge features to determine the amount of information that a target residue gathers from another residue. For residue pairs that are not directly connected by an edge, a learnable pseudo-edge feature is used for attention calculation. Each layer learns a separate pseudo-edge feature that is shared by all non-existing edges. The attention score is then used to weight and sum up node and edge features to produce updated node features. Edge features are also updated by the new node features. Finally, the model generates the sequence from the node features from the last layer in one shot. This memory-efficient global graph attention layer allows for global residue attention while eliminating the need for fully-connected graph construction by learning pseudo-edge features. Residues are able to leverage global interactions and whole-structure features.

Our experiment results demonstrate that our method is effective in handling both entire sequence design and partial sequence design settings. In particular, we validated ProRefiner on the task of single-point mutant design of Transposon-associated transposase B, as a special case of partial sequence design where only one residue can be modified. The proposed ProRefiner successfully identified six variants with improved gene editing activity out of the 20 mutants recommended by the model.

## Results

ProRefiner is trained on CATH v4.2 training set containing 18,204 structures[18].

### Entire Sequence Design

ProRefiner can serve as an add-on module to refine the sequences designed by existing base models. We demonstrate this application on entire sequence design. We experiment with the recent inverse protein folding models as follows.

- GVP-GNN[19] trained on the same training set as ProRefiner. We use the official codebase and default parameters provided by[19] to train and evaluate the model.
- ProteinMPNN[20] trained on selected PDB structures clustered into 25,361 clusters. We use the 48 edges, 0.20 Å noise version of pretrained model weights.

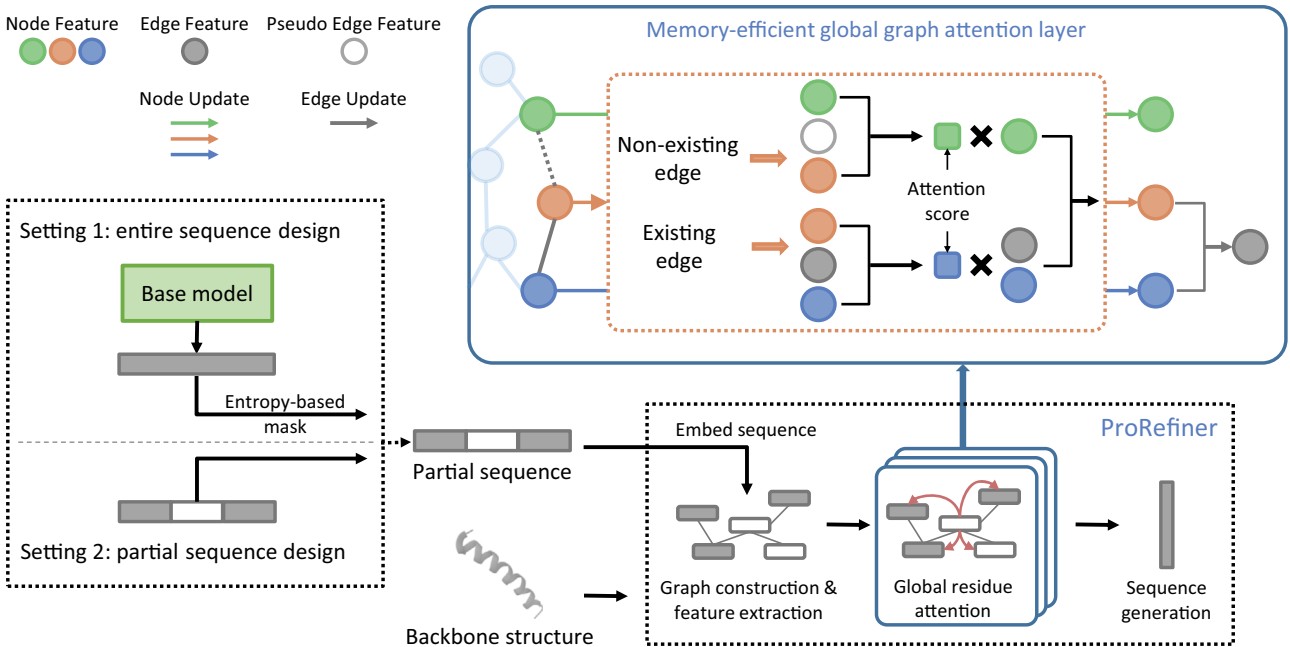

**Fig. 2 | The model architecture of ProRefiner.** A partial sequence, either given or constructed from a base model's generation, and the backbone structure are encoded to obtain the graph features. Several memory-efficient global graph attention layers are employed to propagate the graph features and learn global residue interactions. Finally the whole sequence is generated in one shot.

**Table 1 | Median sequence recovery rates and nssr scores of ProRefiner with different base models on three benchmarks**

| | CATH | | TS50 | | Latest PDB | |
|---|---|---|---|---|---|---|
| | *n* = 1120 | | *n* = 50 | | *n* = 1975 | |
| | Recovery% | nssr% | Recovery% | nssr% | Recovery% | nssr% |
| GVP-GNN | 41.27[40.63, 41.75] | 60.81[60.24, 61.54] | 44.02[41.60, 47.49] | 63.59[61.80, 66.30] | 48.02[47.67, 48.19] | 66.23[65.92, 66.44] |
| ProRefiner+GVP-GNN | 49.89[48.96, 50.29] | 67.93[67.33, 68.44] | 53.75[48.49, 56.86] | 69.33[68.11, 73.33] | 57.77[57.43, 58.11] | 74.18[73.91, 74.42] |
| ProteinMPNN | 42.22[41.49, 43.03] | 60.56[60.00, 61.18] | 43.88[42.21, 46.22] | 61.44[59.72, 63.58] | 49.62[49.31, 49.85] | 66.45[66.17, 66.67] |
| ProRefiner+ProteinMPNN | 51.14[50.44, 52.14] | 69.05[68.42, 69.54] | 53.66[52.24, 56.93] | 71.22[69.49, 73.44] | 59.30[58.94, 59.63] | 75.26[75.11, 75.58] |
| ProteinMPNN-C | 44.94[44.26, 45.70] | 63.79[63.25, 64.29] | 49.05[45.83, 52.42] | 67.87[65.00, 69.70] | 55.34[54.98, 55.70] | 71.52[71.23, 71.76] |
| ProRefiner+ProteinMPNN-C | 50.82[50.00, 51.55] | 69.06[68.50, 69.64] | 54.46[50.93, 57.99] | 71.43[70.20, 73.74] | 60.42[60.05, 60.70] | 75.88[75.63, 76.07] |
| ESM-IF1 | 55.25[54.32, 56.14] | 71.56[70.75, 72.36] | 55.78[52.39, 58.43] | 72.02[69.74, 73.81] | 63.20[62.78, 63.53] | 77.33[76.92, 77.69] |
| ProRefiner+ESM-IF1 | 57.84[57.04, 58.48] | 74.11[73.48, 74.64] | 57.81[55.50, 62.30] | 75.25[71.97, 77.55] | 65.69[65.26, 66.06] | 79.66[79.34, 80.00] |

Data in brackets reports the 95% confidence interval of the median, estimated from 10,000 bootstrap samples.

- ProteinMPNN-C with the same architecture as ProteinMPNN but trained on the same training set as ProRefiner for fair comparison.
- ESM-IF1[21] trained on CATH v4.3 training set with 16,153 structures and 12 million additional structures predicted by Alphafold2[13]. We use the pretrained model weights released by the official codebase.

We conduct experiments on the following three benchmarks.
- CATH. CATH v4.2 dataset[18] is a standard dataset for IPF training and evaluation. We evaluate on its test split of 1120 structures.
- TS50. TS50 is a benchmark set of 50 protein chains proposed by[17]. It has been used by a number of previous works[15,32,33]. There are four structures shared between TS50 and CATH.
- Latest PDB. We collect the latest published structures in PDB to validate the model's ability to generalize to new structures. We select protein structures released after 01/01/2022 with a single chain of length less than 500 and resolution < 2.5 Å, resulting in

1975 protein structures. There are no structures that overlap between Latest PDB and the other two benchmarks.

We report two metrics on all benchmarks: sequence recovery and native sequence similarity recovery (nssr)[34]. A pair of residues is considered similar and contributes to the nssr score if their BLOSUM62 score[35] > 0. Compared with recovery which only considers residue identity, nssr takes residue similarity into account and provides a more specific comparison between two sequences. Additionally, we report perplexity metric in Supplementary Table 1.

In Table 1, we report the median recovery rates and nssr scores of ProRefiner with different base models in comparison to the base model themselves. Among all the base models, ESM-IF1 achieves the best performance, highlighting the effectiveness of data augmentation. ProRefiner consistently achieves high recovery rates and nssr scores even with relatively poor base models, demonstrating its ability to refine the input partial sequences. Additionally, when partial sequences with higher quality are available, ProRefiner outperforms

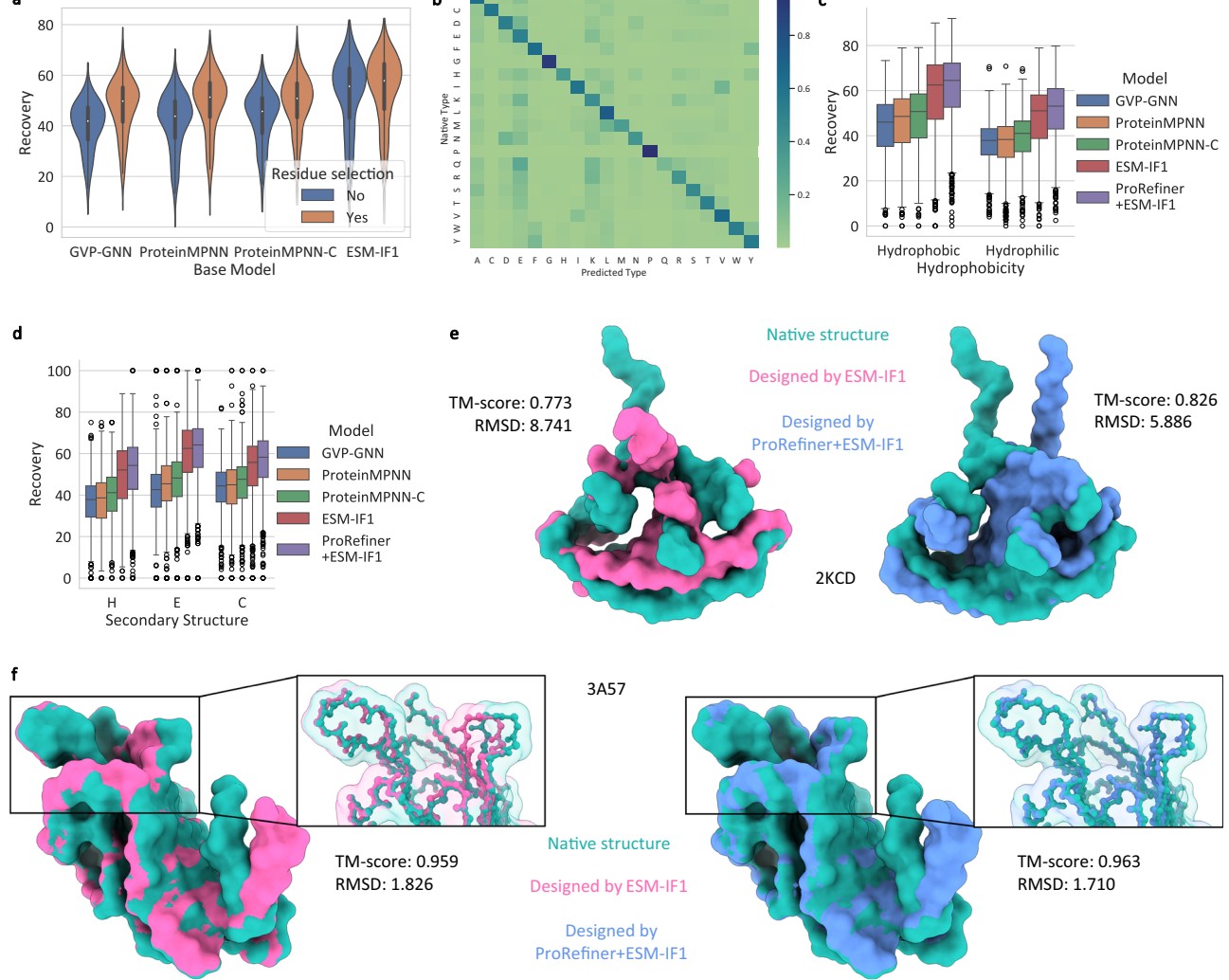

**Fig. 3 | Model performance analysis on entire sequence design. a** Sequence recovery with and without entropy-based residue selection on CATH dataset ($n$ = 1120 structures). The inner box plots show the first quartile, median and the third quartile. Whiskers in box plots extend to the most extreme data point that lies within 1.5 times the inter-quartile range (IQR) from the nearest quartile. **b** Confusion matrix of ProRefiner + ESM-IF1 on CATH dataset ($n$ = 1120 structures). **c, d** Sequence recovery breakdown to hydrophilic and hydrophobic residues and residues on different secondary structures on CATH dataset ($n$ = 1120 structures). The box plots show the first quartile, median and the third quartile. Whiskers in box plots extend to the most extreme data point that lies within 1.5 times IQR from the nearest quartile. **e, f** The predicted structures of sequences designed by ProRefiner + ESM-IF1 and ESM-IF1 for two structures (2KCD and 3A57), compared to the native ones. Alphafold2 is employed to fold the designed sequences. Source data are provided as a Source Data file.

the ESM-IF1 model which uses 12 million additional structures for augmentation. Entropy-based residue selection is used to remove noise in base models' predictions for the subsequent refinement by ProRefiner. To validate its effectiveness, we remove the selection operation and send the entire predicted sequence to ProRefiner. Resulted recovery rate on CATH is plotted in Fig. 3a. Removing entropy-based selection results in a large drop in sequence recovery especially when the base model's recovery is low. This result supports the idea that the noise in the input residue context can significantly limit sequence generation quality and lead to suboptimal sequence designs. Using entropy-based selection to filter out low-quality residue predictions is an effective strategy for improving sequence recovery. The fact that the recovery drop is more pronounced when the base model's recovery is low suggests that the selection operation is particularly important when the input sequence information is less reliable.

In Fig. 3b, we show the confusion matrix of ProRefiner + ESM-IF1 on CATH. A darker cell means a larger portion of the residues of the native type is predicted to be the corresponding type on the horizontal

axis. It can be observed that the residue types the model tends to confuse are also physicochemically similar types, such as ILE vs VAL and GLU vs LYS. Two-sided Mantel test[36] also shows that the confusion matrix is highly correlated with the BLOSUM6 amino acid substation matrices[35] ($p$ value = 0.0001). In Fig. 3c and d, we break down the sequence recovery on CATH to different amino acid types and secondary structures (H stands for $3_{10}$ helix, $\alpha$ helix and $\pi$ helix, E for isolated beta-bridge residue and strand, and C for bend, turn and coil). Our method shows improvement in the recovery of both hydrophilic and hydrophobic residues, with slightly greater improvement seen for residues located on bends, turns or coils.

We assess whether the designed sequences can fold into the target backbones by predicting their structures with Alphafold2. Fig. 3e and f show the results on proteins with PDB codes 2KCD (https://doi.org/10.2210/pdb2KCD/pdb) and 3A57 (https://doi.org/10.2210/pdb3A57/pdb) respectively. We observe that the sequences refined by ProRefiner are predicted to fold into structures more similar to native ones than those designed by ESM-IF1, as evidenced by higher TM-scores[37] and lower root-mean-square deviation (RMSD).

## Partial sequence design

For partial sequence design, ProRefiner fills in the unknown residues based on partial input without the need for base models. To ensure a fair comparison, we evaluate ProRefiner alongside GVP-GNN and ProteinMPNN-C, which are trained on the identical training set used for ProRefiner. Both models are autoregressive models. They implement partial sequence design by replacing the decoded residues with the given fixed amino acids when available during the autoregressive decoding. Additional results for ESM-IF1 model on partial sequence design can be found in Supplementary Table 2. We evaluate the models on the following two benchmarks.

- EnzBench. EnzBench is a standard sequence recovery benchmark consisting of 51 proteins[38]. Designing algorithms are required to recover the native residues on protein design shells with other residues fixed. This benchmark is designed to test the algorithm's ability to model protein binding and overall stability.
- BR_EnzBench. BR_EnzBench[34] aims to test the algorithm's ability to remodel the chosen protein structure. It randomly selects 16 proteins from EnzBench and uses the Backrub server[39] to create an ensemble of 20 near-native conformations for each protein. To further increase the designing difficulty, all residues on the design shell are mutated to alanine, and conformations are then energy-minimized.

When evaluated on EnzBench and BR_EnzBench, identities of residues not on design shells are fixed and available to models. Recovery rates and nssr scores for residues on design shells are reported in Table 2. ProRefiner achieves the highest recovery and nssr on both benchmarks. We further analyze the recovery rate on EnzBench for different amino acids and secondary structures, as shown in Fig. 4a and b. ProRefiner significantly improves the recovery of both hydrophobic and hydrophilic residues and surpasses other models on all secondary structures. We employ Alphafold2 to fold the designed sequences and compare the recovered design shell structures. Fig. 4c shows the designed structures of 1Y52 (https://doi.org/10.2210/pdb1Y52/pdb) and 1Y2U (https://doi.org/10.2210/pdb1Y2U/pdb), which have 19 and 13 designable residues respectively. Design shell residues are shown by atoms. ProRefiner better recovers the design shell structures, with higher TM-scores and lower RMSD. More results and discussion on structure recovery can be found in Supplementary Table 3–5.

## Application on transposon-associated transposase B

Transposon-associated transposase B (TnpB) is thought to be the ancestor of Cas12, the type V CRISPR effector[40,41]. TnpB (408 amino acids) in the D.radiodurans ISDra2 element has been demonstrated to function as a hypercompact programmable RNA-guided DNA endonuclease[42], and its miniature size is suitable for adeno-associated virus-based delivery. However, TnpB exhibits moderate gene editing activity in mammalian cells, limiting its therapeutic application.

We aim to improve the editing activity of TnpB through the design of single-point mutations. We consider the design of a single-point mutation as a partial sequence design, where only one residue is designable, and all others are fixed. With the empirical intuition that a more positively charged surface might potentially improve activity, we restrict the mutation target to the most positively charged amino acid, arginine (R), and restrict the candidate mutation sites to surface residues. We leverage sequence design models to compute a quality score for every candidate site, as illustrated in Fig. 5a. For each site to be examined, we mask this site in the native sequence to get the input partial sequence, and the input backbone structure is the wild-type backbone predicted by Alphafold2[13]. The model then predicts the identity of the masked site in the form of a probability distribution over all amino acid types. We expect that a model that can effectively learn residue interactions will tend to give a higher probability to the

types that are more compatible with the given residue context. Hence, we take the predicted probability for R as a measure of mutant stability. Furthermore, we consider the distance between the $C_\alpha$ of the site and the center of the predicted binding site[43], as empirically mutation sites close to the binding site are more likely to bring improvements. The two scores are combined to obtain a quality score measuring how likely mutation sites can yield stable and improved mutants. All candidate sites are ranked by their quality score and the top 20 are taken as the recommended mutation points.

Our proposed ProRefiner and ProteinMPNN-C are employed for mutant design following the above procedure. 20 mutation points recommended by two models are displayed in Fig. 5b. To test TnpB variants activity in human cells (HEK293T), plasmids encoding the TnpB variants fused with N- and C-terminal nuclear localization (NLS) sequences and reRNA construct targeting a EMX1 site in human genomic DNA (gDNA) were transiently transfected into HEK293T cells. After 96h, gDNA was extracted and analysed by sequencing for the presence of insertions and deletions (Indels) at the targeted cleavage sites. CRISPResso2 is used to analyse Indels, with parameters as follows: minimum of 80% homology for alignment to the amplicon sequence, quantification window of 20 bp and ignoring substitutions to avoid false positives. Experiments show that 6 arginine substitutions designed by ProRefiner exhibit above 1.2-fold improvement in indel activity relative to TnpB WT, and 3 by ProteinMPNN-C. Results are given in Fig. 5c. Additionally, off-target of the variant with the highest activity, TnpB K84R, is compared with the TnpB WT. As shown in Fig. 5d, the increase in activity leads to a degree of non-specific cleavage as expected, which may compromise the nuclease's specificity.

This experiment demonstrates that the proposed ProRefiner is effective at modeling residue interactions within a structural environment and generating sequences that best fit a given 3D context. It can be used in combination with other property measures to redesign existing proteins and improve their stability or other qualities that depend on protein stability.

## Discussion on model designs

We examine several key designs in ProRefiner model architecture. Our investigation revealed that the introduced global attention layers could learn and exploit meaningful residue interactions. Specifically, we identity the residues to which a target residue pays the most attention, indicated by the highest attention scores. It's observed that many important residue interactions are well learned and represented by the attention operation. Fig. 6a–c provide examples for three typical chemical bonds in protein structures, where the target residues are shown in blue, and the three residues with the highest mean attention scores are shown in orange. For example, in the case of HIS 9 on 2KCD (https://doi.org/10.2210/pdb2KCD/pdb), LEU 5 is among its most attended residues. HIS 9 forms a hydrogen bond with LEU 5 on the $\alpha$ helix and this interaction is well learned by the attention layers. Similarly, ILE 70 on the sheet heavily attends to ASN 54, which it forms a hydrogen bond with. For T4-lysozyme (https://doi.org/10.2210/pdb1LYD/pdb), ASP 70 forms a hydrogen bond with LEU 66 and a salt bridge with HIS 31, and both residues are among its most attended residues. The attention operation also captures a disulfide bond between CYS 99 and CYS 94 of human Ero1-alpha (https://www.uniprot.org/uniprotkb/Q96HE7/entry).

We further validate the effectiveness of two key components in ProRefiner: global attention mechanism and partial sequence input. Two models are trained for ablation study: (1) a model without the global attention view, where residues only attend to their graph neighbors, and pseudo edge features are therefore not used; and (2) a model without partial sequence input during training, where all residues are masked. We compare the recovery rates of ablated models on CATH (using base model ESM-IF1), EnzBench and BR_EnzBench. Results are given in Table 3. It is observed that removing either

**Table 2 | Median sequence recovery rates and nssr scores on EnzBench and BR_EnzBench**

| | EnzBench | | BR_EnzBench | |
| | n = 51 | | n = 320 | |
| | Recovery% | nssr% | Recovery% | nssr% |
|---|---|---|---|---|
| GVP-GNN | 41.38[36.36, 42.86] | 57.89[55.00, 63.16] | 29.41[27.27, 31.58] | 47.83[47.37, 52.17] |
| ProteinMPNN-C | 52.00[50.00, 59.09] | 70.00[65.00, 77.78] | 40.91[40.00, 42.48] | 60.00[59.09, 60.87] |
| ProRefiner | 57.89[55.00, 63.64] | 73.68[70.59, 78.26] | 43.48[41.64, 44.44] | 60.87[59.09, 63.64] |

Data in brackets reports the 95% confidence interval of the median, estimated from 10,000 bootstrap samples.

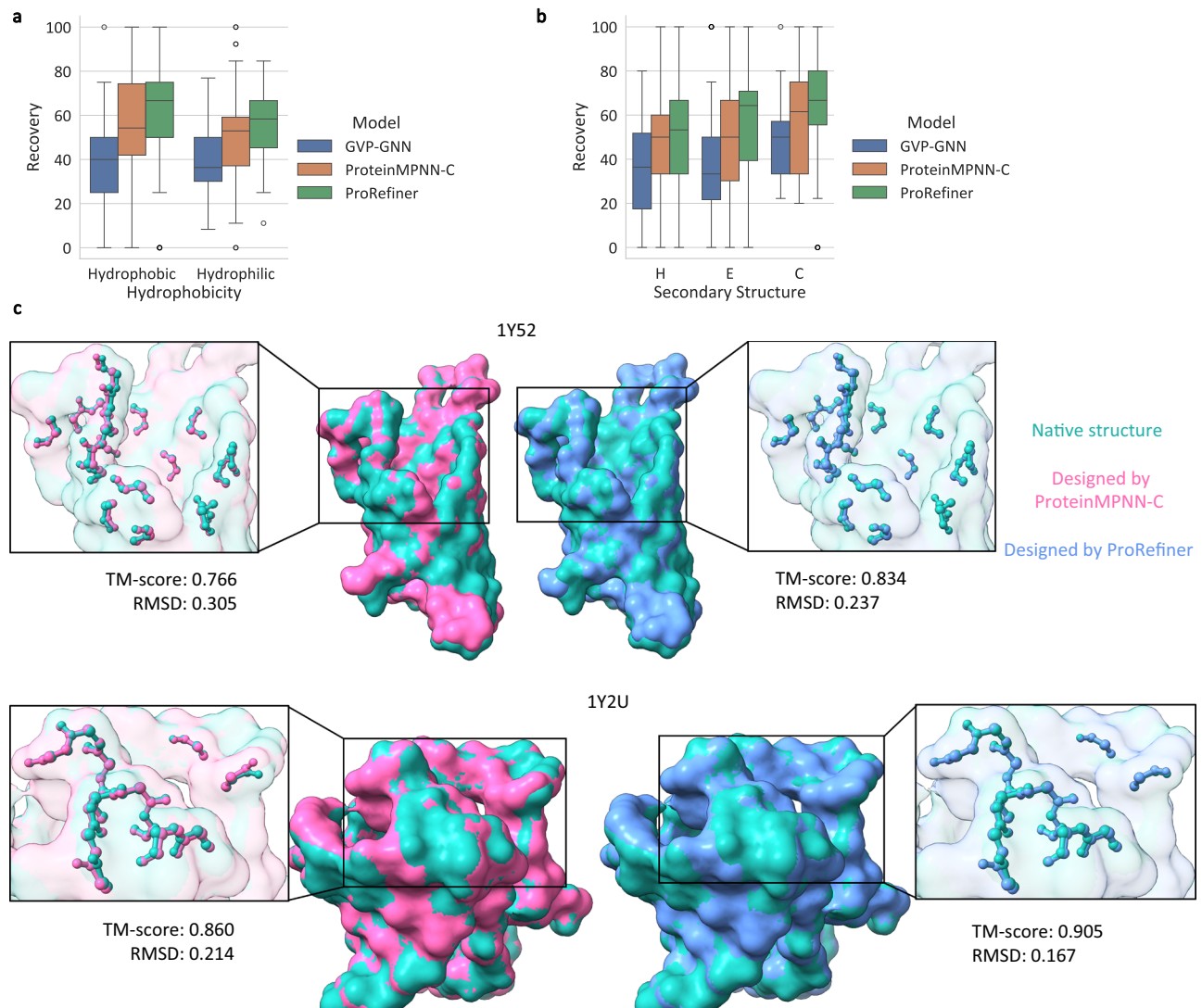

**Fig. 4 | Model performance analysis on partial sequence design. a, b** Sequence recovery breakdown to hydrophilic and hydrophobic residues and residues on different secondary structures on EnzBench dataset (n = 51 structures). The box plots show the first quartile, median and the third quartile. Whiskers in box plots extend to the most extreme data point that lies within 1.5 times IQR from the nearest quartile. **c** The predicted structures of sequences designed by ProRefiner and ProteinMPNN-C for two structures (1Y52 and 1Y2U), compared to the native ones. Design shells are plotted in atoms. Alphafold2 is employed to fold the designed sequences. Source data are provided as a Source Data file.

component results in a drop in model performance and we get *p* value < 0.05 by paired two-sided t-test on all datasets, indicating a significant improvement of introducing the global attention view and input partial sequence. Notably, the model without partial sequence input exhibits a significantly larger performance degradation on BR_Enz-Bench, indicating that when input structures are not accurate, the ability to utilize sequence information becomes more important. We

also test the robustness of these models to the input residue noise. For sequences generated by base model ESM-IF1, we mask different percentages of residues with the highest entropy, and plot the model's median recovery rate on the CATH benchmark in Fig. 6d. The model trained without partial sequence cannot really utilize input partial sequence, and thus exhibits relatively stable performance. The recovery rate of the other two models first increases as more noisy

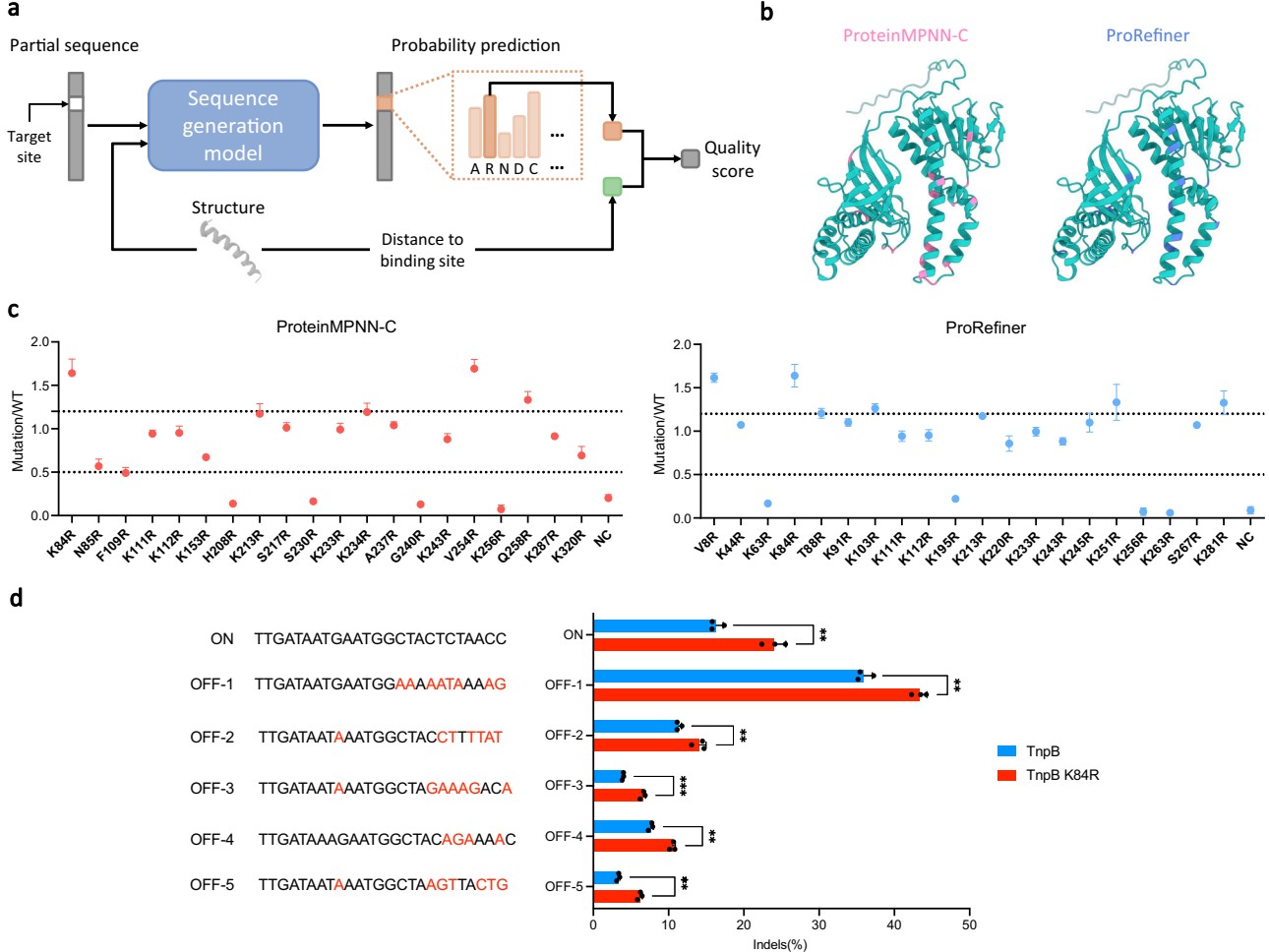

**Fig. 5 | The procedure and results of TnpB single-point mutation design. a** The process of computing the quality score of one target mutation site on TnpB WT sequence. **b** The mutation sites recommended by ProteinMPNN-C and ProRefiner are marked in red and blue respectively. **c** The improvement of variants recommended by ProteinMPNN-C and ProRefiner in indel activity relative to TnpB WT. All statistical analysis were performed on $n$ = 3 biologically independent experiments and data is shown as the mean ± SD of three biological replicates. **d** Indel formation at the on-target and off-target sites observed for TnpB WT and TnpB K84R. Off-target sites are chosen following[50]. All statistical analysis were performed on $n$ = 3 biologically independent experiments and data is shown as the mean ± SD of three biological replicates with actual values overlaid. $P$ values were derived by a two-sided Student's t-test with ** denoting $P < 0.01$ and *** $P < 0.001$. The exact P-values from top to bottom are 0.00188, 0.00104, 0.00792, 0.000214, 0.000508 and 0.000207. Source data are provided as a Source Data file.

residues are removed, and drops slightly when too many residues are masked and less sequence information is left. The model with a global attention view consistently outperforms the model with local view, demonstrating a better ability to leverage the input residue context, as residue information is available to every node, not just the ones close to them.

Finally, we compare the performance of the proposed memory-efficient global attention layers against the original global attention layers. We use ProRefiner - PsFeat to denote the model that uses the vanilla global attention layers without learning pseudo-edge features. Figure 6e illustrates the runtime and GPU memory utilization of the two models. The measurements are obtained by 16 independent runs. ProRefiner exhibits linear time and memory complexity, whereas ProRefiner - PsFeat introduces quadratic complexity due to the construction of fully connected graphs. The predictive performance of the two models is presented in Fig. 6f. We experiment on CATH benchmark with base model ESM-IF1 for entire sequence design and EnzBench and BR_EnzBench benchmarks for partial sequence design. The results indicate that while ProRefiner shows slightly greater performance variance on certain benchmarks, its overall performance remains similar and comparable to that of ProRefiner - PsFeat. Additionally, we compare the predictive performance of the two models on other tasks and report the results in Supplementary Table 6.

## Discussion

In this work, we attempt to take a step towards better modeling and learning of inter-body interactions within protein structures, by proposing a method for inverse protein folding. We develop a two-pronged approach that incorporates a residue selection technique and a memory-efficient global graph attention model, which work jointly to achieve effective selection and utilization of high-quality residue interactions. Our experiments demonstrate that the proposed ProRefiner is able to capture meaningful inter-residue bonds and achieve high sequence recovery on several protein design benchmarks. We also apply the model to redesign TnpB and successfully discovered six mutants with enhanced editing activity. Our results highlight the potential of our method to facilitate the design of proteins with improved functional properties. Additionally, the memory-efficient graph attention module proposed herein provides an efficient means of modeling graph-structured data where global dependencies are critical. Potential future research directions could involve the application of this module to other protein-related tasks and the examination of other biomolecules.

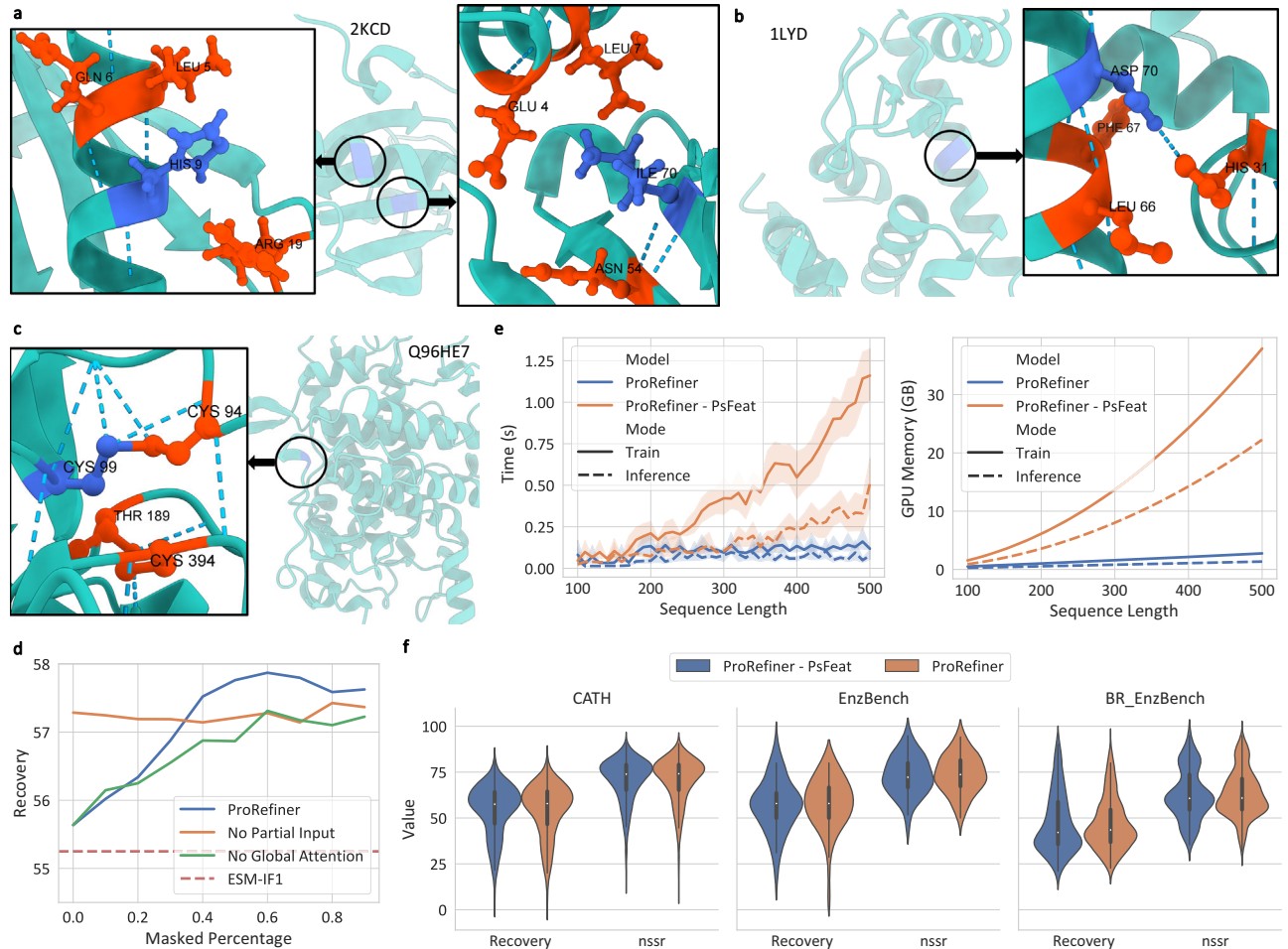

**Fig. 6 | Analysis on model design and ablation study. a** Hydrogen bonds between two target residues in blue (HIS 9, ILE 70 on 2KCD) and their most attended residues in orange. **b** A salt bridge and a hydrogen bonds between ASP 70 of T4-lysozyme and its most attended residues. **c.** A disulfide bond between CYS 99 and CYS 94 of human Ero1-alpha (Q96HE7). **d** Models' median recovery rate on CATH dataset ($n$ = 1120 structures), with respect to different percentages of residues masked on sequences generated by base model ESM-IF1. **e** Runtime and GPU memory usage of ProRefiner and the model without pseudo edge feature, denoted by ProRefiner - PsFeat. Data is obtained through 16 independent runs and the plots show the mean with the 95% confidence interval of the mean, estimated from 1000 bootstrap samples. **f** Recovery rates and nssr scores of ProRefiner and ProRefiner - PsFeat on CATH ($n$ = 1120), EnzBench ($n$ = 51) and BR_EnzBench ($n$ = 320). ESM-IF1 is used as the base model for inference on CATH. The inner box plots show the first quartile, median and the third quartile. Whiskers in box plots extend to the most extreme data point that lies within 1.5 times IQR from the nearest quartile. Source data are provided as a Source Data file.

**Table 3 | Median recovery of ProRefiner (the last row) and two ablated models, either without global attention view or without partial sequence input**

| Model | | CATH | | EnzBench | | BR_EnzBench | |
|---|---|---|---|---|---|---|---|
| | | $n$ = 1120 | | $n$ = 51 | | $n$ = 320 | |
| Global Attention | Partial Input | Recovery% | $p$ value | Recovery% | $p$ value | Recovery% | $p$ value |
| | ✓ | 57.26[56.34, 58.06] | 1.6e-5 | 55.56[52.63, 62.50] | 0.0397 | 42.86[40.91, 45.00] | 0.0259 |
| ✓ | | 57.21[56.41, 57.96] | 0.0023 | 56.52[50.00, 60.71] | 0.0137 | 35.71[35.00, 36.84] | 2.1e-16 |
| ✓ | ✓ | 57.84[57.04, 58.48] | N/A | 57.89[55.00, 63.64] | N/A | 43.48[41.64, 44.44] | N/A |

The base model to produce the results on CATH is ESM-IF1. Data in brackets reports the 95% confidence interval of the median, estimated from 10,000 bootstrap samples. The $p$ values when comparing two ablated models to ProRefiner are reported.

## Methods
### Graph representation of proteins
A protein structure is represented as a proximity graph $\mathcal{G} = (\mathcal{V}, \mathcal{E})$, where $\mathcal{V} = \{v_1, v_2, \ldots, v_N\}$ denotes the residue nodes and $\mathcal{E} = \{e_{ij}\}$ denotes the directed edges form $v_j$ to $v_i$, where residue $v_j$ is among the $k$ = 30 nearest neighbors of $v_i$ in terms of $C_\alpha$ distance. Each node $v_i$ has the following structural features:
- sin and cos value of dihedral angles;

- unit vectors from the previous and next residues on sequence to $v_i$ in terms of $C_\alpha$ position.

Each edge $e_{ij}$ has the following features:
- Gaussian radial basis functions encoding of interatomic distances between $N, C_\alpha, C, O$ and a virtual $C_\beta$, and encoding of distance on sequence $i - j$[20];
- unit vector from $v_j$ to $v_i$ in terms of $C_\alpha$ position.

We then employ 2 geometric vector perceptrons layers[19] to embed the structural features.

For sequence features, at training time, we randomly mask 70% of residues in the native sequences. Among the remaining 30% of the residues whose identity is available to the model, we randomly select 3% and replace their identity with random amino acid types. During inference, the input partial sequences are provided as-is without any masks or replacements. The partial sequences are then embedded as node features. Masked residues with unknown identity are treated as a special type unknown. The sequence node features are concatenated with structural node features. Resulting node features are denoted as $\mathbf{H^0} \in \mathbb{R}^{N \times d}$ where $\mathbf{h_i^0} \in \mathbb{R}^d$ denotes the feature of $v_i$. Resulting edge features are denoted as $\mathbf{E^0} \in \mathbb{R}^{N \times k \times d}$ where $\mathbf{E_i^0} \in \mathbb{R}^{k \times d}$ is the features of $k$ neighbors of $v_i$ and $\mathbf{e_{ij}^0} \in \mathbb{R}^d$ denotes the feature of edge from $v_j$ to $v_i$.

## Memory-efficient global graph attention model

Attention is first introduced in Transformer model[26]. Let $\mathbf{H} \in \mathbb{R}^{N \times d}$ denote the $d$-dimension features of the input sequence with length $N$. The self-attention module updates the input features according to the following equations:

$$\text{SelfAtten}\,(\mathbf{H}) = \mathbf{AV}, \tag{1}$$

$$\mathbf{A} = \text{Softmax}\left(\frac{\mathbf{QK^T}}{\sqrt{d_K}}\right), \tag{2}$$

$$\mathbf{K} = \mathbf{HW_K}, \mathbf{Q} = \mathbf{HW_Q}, \mathbf{V} = \mathbf{HW_V}, \tag{3}$$

where $\mathbf{W_K} \in \mathbb{R}^{d \times d_K}, \mathbf{W_Q} \in \mathbb{R}^{d \times d_K}, \mathbf{W_V} \in \mathbb{R}^{d \times d_V}$ are parameters to map $\mathbf{H}$ to keys, queries and values.

There have been attempts to employ Transformer architecture for learning on graphs, with nodes denoted as sequence tokens. To utilize the global view provided by the original self-attention on graphs with edge features, previous works generally incorporate edge features into the attention matrix $\mathbf{A}$:

$$\mathbf{A} = \text{Softmax}\left(f\left(\frac{\mathbf{QK^T}}{\sqrt{d_K}}, \phi(\mathbf{E})\right)\right), \tag{4}$$

where $\mathbf{E} \in \mathbb{R}^{N \times N \times d}$ is the $d$-dimension edge features between each pair of nodes, $\phi$ estimates the correlations of node pairs from edge features, which could be linear transformation[27] or more sophisticated functions[31], and $f$ is an aggregation function, which could be element-wise addition[27,31] or multiplication[27]. These methods have two limitations. First, to construct the edge feature matrix $\mathbf{E}$, they require fully connected graphs as input and the memory complexity will be $\mathcal{O}(N^2)$. Second, the edge features are not fully leveraged. They are only involved in attention computation and can not be used to update node features or vice versa.

ProRefiner model is composed of a stack of $L$ memory-efficient global graph attention layers. In each layer, nodes can globally attend to all other nodes and edge features between node pairs serve as additive attention bias terms. For non-existing edges, one solution is to convert arbitrary graphs to fully connected graphs before entering the model, then Equation (4) could be used. This could be done by setting $k$ to a large enough number or using a fixed masking value/vector for non-existing edges as in previous works[27]. This operation increases memory complexity from $\mathcal{O}(N \times k)$ to $\mathcal{O}(N^2)$. To avoid the conversion, we design a learnable pseudo edge feature in each layer. Let $\mathbf{H^l}$ and $\mathbf{E^l}$ denote the input features of the $l$th layer. The attention is computed as

follows:

$$\mathbf{A^l} = \text{Softmax}\left(\frac{\mathbf{Q^l}(\mathbf{K^l})^{\mathbf{T}}}{\sqrt{d}} + \mathbf{B^l}\right), \tag{5}$$

$$\mathbf{B_{ij}^l} = \begin{cases} (\mathbf{w_B^l})^{\mathbf{T}}\mathbf{e_{ij}^l} & j \in \mathcal{N}_i \\ (\mathbf{w_B^l})^{\mathbf{T}}\boldsymbol{\beta}^{\mathbf{l}} & j \notin \mathcal{N}_i \end{cases}, \tag{6}$$

$$\mathbf{K^l} = \mathbf{H^l W_K^l}, \mathbf{Q^l} = \mathbf{H^l W_Q^l}, \tag{7}$$

where $\mathbf{B^l}$ is the attention bias, $\mathbf{W_K^l} \in \mathbb{R}^{d \times d}, \mathbf{W_Q^l} \in \mathbb{R}^{d \times d}, \mathbf{w_B^l} \in \mathbb{R}^d$ are parameters, $\boldsymbol{\beta}^{\mathbf{l}} \in \mathbb{R}^d$ is the pseudo edge feature in layer $l$. Learning a pseudo edge feature for each layer is more adaptive and flexible than using one fixed masking value across all layers and provides a better approximation to using fully connected graphs.

The attention score $\mathbf{A^l}$ is then used to aggregate node features as well as edge features. Node features are weighted and summed as in the vanilla self-attention. Edge features are aggregated with normalized weights and concatenated with aggregated node features. Finally, a linear layer is employed to map the concatenated feature to dimension $d$:

$$\hat{\mathbf{h}}_\mathbf{i}^\mathbf{l} = \left[\sum_j \mathbf{A_{ij}^l V_j^l} \,\|\, \sum_{j \in \mathcal{N}_i} \frac{\mathbf{A_{ij}^l}}{\gamma_i^l} \mathbf{e_{ij}^l}\right] \mathbf{W_N^l}, \tag{8}$$

$$\mathbf{V^l} = \mathbf{H^l W_V^l}, \tag{9}$$

where $\mathbf{W_V^l} \in \mathbb{R}^{d \times d}, \mathbf{W_N^l} \in \mathbb{R}^{2d \times d}$ are parameters, $\|$ means concatenation operation, and $\gamma_i^l$ is a normalization term to normalize the sum of edge weights to 1.

Then a residue connection and layer normalization are adopted to output the final updated node features:

$$\mathbf{H^{l+1}} = \text{LayerNorm}\,(\hat{\mathbf{H}}^\mathbf{l} + \mathbf{H^l}). \tag{10}$$

The edge features will then be updated as follows, with $\mathbf{W_E^l} \in \mathbb{R}^{3d \times d}$ being parameters:

$$\hat{\mathbf{e}}_{\mathbf{ij}}^\mathbf{l} = [\mathbf{h_i^{l+1}} \,\|\, \mathbf{e_{ij}^l} \,\|\, \mathbf{h_j^{l+1}}]\mathbf{W_E^l}, \tag{11}$$

$$\mathbf{E^{l+1}} = \text{LayerNorm}\,(\hat{\mathbf{E}}^\mathbf{l} + \mathbf{E^l}). \tag{12}$$

Note that we adopt this naive edge feature update here for its empirical effectiveness and implementation simplicity. However, it cannot ensure the triangle inequality on distances[13]. Incorporating more sophisticated edge update method for triangle inequality constraints could be a promising future direction.

To leverage edge features under the global attention mechanism, compared with $\mathcal{O}(N^2)$ by previous works, our memory-efficient global graph attention only needs $\mathcal{O}(N \times k + L)$ additional memory, and therefore allows designing longer sequences.

The output node features from the last layer will be mapped to the distribution over 20 residue types through a linear layer with parameter $\mathbf{W_P} \in \mathbb{R}^{d \times 20}$:

$$\mathbf{p_i} = \text{Softmax}\left(\mathbf{h_i^L W_P}\right). \tag{13}$$

Negative log-likelihood loss is used during training.

## Entire sequence design with base model

In entire sequence design setting, we use an entropy-based residue selection method to construct the partial input sequence. Suppose $\mathbf{p_i^b}$

**Table 4 | The barcoded primers used in PCR1**

| | |
|---|---|
| EMX1-PCR-seq-F1 | ACACTCTTTCCCTACACGACGCTCTTCCGATCTGAGggtggttcaggcctccttcccac |
| EMX1-PCR-seq-F2 | ACACTCTTTCCCTACACGACGCTCTTCCGATCTTAAggtggttcaggcctccttcccac |
| EMX1-PCR-seq-F3 | ACACTCTTTCCCTACACGACGCTCTTCCGATCTTCAggtggttcaggcctccttcccac |
| EMX1-PCR-seq-F4 | ACACTCTTTCCCTACACGACGCTCTTCCGATCTTCGggtggttcaggcctccttcccac |
| EMX1-PCR-seq-F5 | ACACTCTTTCCCTACACGACGCTCTTCCGATCTTGCggtggttcaggcctccttcccac |
| EMX1-PCR-seq-R1 | GTGACTGGAGTTCAGACGTGTGCTCTTCCGATCTACCcaagatgctaagtgatgacagg |
| EMX1-PCR-seq-R2 | GTGACTGGAGTTCAGACGTGTGCTCTTCCGATCTCAGcaagatgctaagtgatgacagg |
| EMX1-PCR-seq-R3 | GTGACTGGAGTTCAGACGTGTGCTCTTCCGATCTCCTcaagatgctaagtgatgacagg |
| EMX1-PCR-seq-R4 | GTGACTGGAGTTCAGACGTGTGCTCTTCCGATCTCTAcaagatgctaagtgatgacagg |
| EMX1-PCR-seq-R5 | GTGACTGGAGTTCAGACGTGTGCTCTTCCGATCTGTAcaagatgctaagtgatgacagg |

is the probability distribution of $v_i$ predicted by a base model. We compute the entropy $en_i^b$ of distribution $\mathbf{p_i^b}$:

$$en_i^b = \mathbb{E}\left[-\log \mathbf{p_i^b}\right] \tag{14}$$

Residues with the least entropy are selected and retained while others are masked. To account for the varying ability of different base models to recover native sequences, we select different percentages of residues depending on the base model being used. The percentage of residues chosen for each base model is determined based on the recovery rate on the validation split of CATH v4.2. We experiment with percentages ranging from 5% to 50% for each base model, and select the percentage resulting in the highest recovery rate. Specifically, we choose 10% for GVP-GNN, 10% for ProteinMPNN, 15% for ProteinMPNN-C, and 35% for ESM-IF1. These percentages are roughly correlated with the sequence recovery performance of each base model.

The partial sequence is fed into ProRefiner to get the probability predictions $\mathbf{p_i}$ with entropy $en_i$. Finally, the predictions from the base model and ProRefiner will be weighted by their entropy and fused together:

$$\hat{\mathbf{p_i}} = \frac{\exp(-en_i)}{\exp(-en_i)+\exp(-en_i^b)}\mathbf{p_i} + \frac{\exp(-en_i^b)}{\exp(-en_i)+\exp(-en_i^b)}\mathbf{p_i^b}. \tag{15}$$

The final predicted residue type will be the argmax of $\hat{\mathbf{p_i}}$.

**Experiment details of TnpB design**

Plasmid vector construction. The TnpB gene was optimized for expression in human cells through codon optimization and the optimized sequence was synthesized for vector construction (Sangon Biotech). The final optimized sequence was inserted into a pST1374 vector, which contained a CMV promoter and nuclear localization signal sequences at both the 5′ and 3′ termini. reRNA sequences were synthesized and cloned into a pGL3-U6 vector. Spacer sequences (EMX1: 5′- ctgtttctcaggatgtttgg -3′) were cloned into by digesting the vectors with BsaI restriction enzyme (New England BioLabs) for 2 h at 37°C. The resulting vector constructs were verified through Sanger sequencing to ensure accuracy.

TnpB engineering. The construction of TnpB mutants was carried out through the use of site-directed mutagenesis. PCR amplifications were performed using Phanta Max Super-Fidelity DNA Polymerase (Vazyme). The PCR products were then ligated using 2X MultiF Seamless Assembly Mix (ABclonal). Ligated products were transformed into DH5α E. coli cells. The success of the mutagenesis was confirmed through Sanger sequencing. The modified plasmid vectors were purified using a TIANpure Midi Plasmid Kit (TIANGEN).

Cell culture and transfection. HEK293T cells were maintained in Dulbecco's modified Eagle medium (Gibco) supplemented with 10% fetal bovine serum (Gemini) and 1% penicillin-streptomycin (Gibco) in an incubator (37°C, 5% $CO_2$). HEK293T cells were transfected at 80% confluency with a density of approximately $1 \times 10^5$ cells per 24-well using ExFect Transfection Reagent (Vazyme). For indel analysis, 500 ng of TnpB plasmid plus 500 ng of reRNA plasmid was transfected into 24-well cells.

DNA extraction and Deep sequencing. The genomic DNA of HEK293T cells was extracted using QuickExtract DNA Extraction Solution (Lucigen). Samples were incubated at 65°C for 60 minutes and 98°C for 2 minutes. The lysate was used as a PCR template. The first round PCR (PCR1) was conducted with barcoded primers to amplify the genomic region of interest using Phanta Max Super-Fidelity DNA Polymerase (Vazyme). The products of PCR1 were pooled in equal moles and purified for the second round of PCR (PCR2). The PCR2 products were amplified using index primers (Vazyme) and purified by gel extraction for sequencing on the Illumina NovaSeq platform. The specific barcoded primers used in PCR1 are listed in Table 4.

**Statistics and reproducibility**

No statistical method was used to predetermine sample size since the methods are evaluated on the full CATH test set, TS50, Latest PDB, EnzBench and BR_EnzBench dataset. We excluded multichain structures and structures of a length more than 500 or resolution > 2.5 Å for Latest PDB dataset. No data were excluded from the analysis for other benchmarks. All genome editing attempts were performed with at least three biological repeats. All attempts at reproducibility were successful and standard deviations were in the expected ranges. The experiment randomization and blinding are not applicable since we are not making a comparison between different groups.

**Reporting summary**

Further information on research design is available in the Nature Portfolio Reporting Summary linked to this article.

# Data availability

All relevant data supporting the key findings of this study are available within the article and its Supplementary Information files. CATH dataset (v4.2) is available at http://people.csail.mit.edu/ingraham/graph-protein-design/data/. TS50 dataset (v2.0) is available at: https://zenodo.org/record/6650679#.ZDJJJNhVByhY[44]. EnzBench is available as part of the standard Rosetta package (v3.13) which could be downloaded from https://www.rosettacommons.org/software/license-and-download with a license. BR_EnzBench is provided by and available from the paper[34]. Latest PDB dataset is available at https://drive.google.com/file/d/1Ate5I0Hz5GwzOJN4sQL_RrDUkjxMJZO0u/view?usp=sharing. Ligand Binding Affinity dataset (v0.1) is available at https://zenodo.org/record/4914718[45]. Small Molecule Properties dataset (v0.1) is available at https://zenodo.org/record/4911142[46]. Source data is available at Figshare (https://doi.org/10.6084/m9.figshare.23913147)[47]. Source data are provided with this paper.

## Code availability

TM-score and RMSD calculation used TM-score v20220415 from https://zhanggroup.org/TM-score/. Mantel test used mantel v2.2.0 from https://github.com/jwcarr/mantel. Other data analysis used Python v3.9.16 (https://www.python.org/), NumPy v1.24.3 (https://numpy.org/), SciPy v1.10.1 (https://scipy.org/), PyTorch v1.13.0 (https://pytorch.org/), pandas v2.0.2 (https://pandas.pydata.org/), Matplotlib v3.7.1 (https://matplotlib.org/), seaborn v0.12.2 (https://seaborn.pydata.org/) and Biopython v1.80 (https://biopython.org/). Structure visualizations were created in ChimeraX v1.6.1 (https://www.cgl.ucsf.edu/chimerax/). Indel analysis used CRISPResso2 (https://github.com/pinellolab/CRISPResso2). The code developed in this manuscript and pretrained model weights are provided through Colab: https://colab.research.google.com/drive/1a6VW-BB0twEwL65sE_dUAM42wdSm6RZp?usp=sharing, Code Ocean: https://codeocean.com/capsule/9492154/tree[48] and Github: https://zenodo.org/records/10030882[49].

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

## Acknowledgements

G.C. acknowledges support from National Key R&D Program of China (2022YFE0200700), National Natural Science Foundation of China (62006219, 62376254) and Natural Science Foundation of Guangdong Province (2022A1515011579). E.W. acknowledges support from National Natural Science Foundation of China (22007082) and Natural Science Foundation of Zhejiang Province, China (LQ21B030013). J.Z. acknowledges support from National Key R&D Program of China (2022YFC2702705). P.A.H. acknowledges support from Hong Kong Innovation and Technology Fund (ITS/170/20, ITS/241/21).

## Author contributions

P.A.H. and G.C. conceived and supervised this work. X.Z. developed the code for the proposed algorithm. X.Z., G.C. and J.Y. designed the in silico experiments and analyzed the experiment data. G.C., E.W., Z.L., X.H. and J.T. designed the TnpB mutation design experiment. X.Z., J.Z. and C.M. conducted the TnpB mutation design experiment and analyzed the data. X.Z., J.Z. and C.M. drafted the manuscript and prepared the supplementary materials. G.C., J.Y., E.W., J.H., X.H. and P.A.H. revised the manuscripts. All authors participated in the discussions and agreed with the contents of this work.

## Competing interests

The authors declare no competing interests.
