## [Peer Review File · Nature Communications]

ProRefiner: An Entropy-based Refining Strategy for Inverse Protein Folding with Global Graph AttentionReviewer #1 (Remarks to the Author):

This paper addresses the problem of predicting amino acid sequences that can lead to specific protein 3D structures. It introduces two interesting ideas, an entropy-based approach for removing 'noise' in the residue features and memory-efficient attention with edge awareness.

First, the scope of memory-efficient graph attention is much broader than the problem of inverse protein folding (IPF.) For example, AlphaFold2's row attention includes a bias term accounting for edge-features scales quadratically and the pair-representation track scales cubically (to account for the triangular inequality). It would be interesting in future works to see how memory-efficient graph attention could be applied to tasks beyond (IPF.)

Despite the interesting ideas introduced in the manuscript, I had difficulty following the flow of the discussion and, more importantly, the exact meaning of 'our model.'

For example, the 'memory-efficient global graph attention mechanism' discussion in the main text, starting on page 4, is not particularly informative. Either it targets readers without prior knowledge about vanilla attention, and I do not think they can follow much of the arguments, or those familiar with the attention mechanism, the discussion is relatively shallow.

Returning to the 'model' in the Results section, I do not understand what is reported in Table 1. On top of reporting the performance of existing models (GVP, ProteinMPNN, and ESM), I would expect one row reporting the model results, which combines the entropy-based approach and the memory-efficient attention graph attention (with and without partial sequences). I do not understand the meaning of 'Ours GVP-GNN,' and the same for ProteinMPNN and ESM. Moreover, what is the meaning of 'Ours' in Table 2?

For example, is 'Ours ESM' trained using the same regimen as the original ESM, using 12M additional structures predicted from AlphaFold2?

On top of recovery and nssr, it would be interesting to report the perplexity. If I understand correctly what the authors mean by noise, this could be related to the perplexity score.

I am unsure of the difference between the results in Table 1 and Fig3.A. I understand that Table.1 reports results about partial sequences and Fig 3.A about whole sequences.

The main results reported in Table 1 should be about the entire sequence case (and not partial sequence), as this is the standard in the literature.

The ablation study assessing the relative importance of global attention and partial input is run by considering on EnzBench and BR_EnzBench. Running such an ablation on CATH would have been more interesting, as it allows direct comparison with existing models.

The global graph attention model updates the node and edge features. I have a comment regarding the edge update on Equation 11. A naïve update like in Eq.11 may violate the triangular inequalities, i.e., edge features that have to fulfill the triangular inequality are not independent. Such a condition is enforced in AlphaFold2 by considering a cubic update; roughly, an edge e_{ij} is updated conditioned on all k : $e_{ik} e_{kj}$. I am curious if the authors thought of such constraints.

Reviewer #2 (Remarks to the Author):

In the manuscript, the authors proposed a refinement model for inverse protein folding, which designs sequences that are consistent with a given backbone structure and partial sequence information. The model utilizes an entropy-based masking strategy to construct partial sequence context for later sequence prediction. In addition, the model introduces pseudo edge features shared by non-existing edges in its graph model, which significantly reduces the memory usage compared to fully-connected graph neural networks. The potential significance of this manuscript

comes from two key aspects: first, improvements in the inverse folding process facilitate structure-based de novo protein design; second, a memory-efficient graph neural network of equivalent model performance offers a powerful mechanism for reasoning with larger proteins.

However, there are two major concerns with the conclusion drawn from this manuscripts:

1. The authors evaluated model performance based on sequence recovery rate and native sequence similarity recovery. However, it is unclear how well these designed sequences fold into the target structures. It is possible that a model with lower sequence recovery rate generates sequence designs better matching with input target structures. Thus, further in-silico structural validations are essential for demonstrating the effectiveness of the proposed refinement method.
2. Memory-efficient global attention layer utilizes a shared pseudo edge feature for non-existing edges in order to save memory. While the authors illustrate the model's memory advantage in Figure 6E, it is crucial to compare the predictive performance between the memory-efficient global attention layer and the original global attention layer (and preferably across several evaluation tasks). Given provided information in the manuscript, it is inconclusive on the effectiveness and usefulness of the proposed memory-efficient global attention layer.

In addition, there are several points that requires further clarifications:

1. Details on how base models are trained and evaluated for both entire sequence design and partial sequence design are needed to better understand the performance of the proposed refinement model. For example, ProteinMPNN, to some degree, exhibits a tradeoff between sequence recovery ratio and in-silico success rate, controlled by the training noise level. Thus, it is important to provide the backbone noise ratio for the ProteinMPNN model used.
2. For full sequence inverse folding, the model relies on base models to generate partial sequence. What is the strength of the entropy-based mask? What is the percentage of residues that are provided as context for the later model?
3. For partial sequence design, is ProteinMPNN retrained with partial sequence provided as input? If not, how are GVP-GNN and ProteinMPNN used for partial sequence design? Moreover, how does ESM perform on partial sequence design?
4. At training time, what percentage of the residues are masked as unknown?
5. How much overlapping is there among the three evaluation datasets?

Reviewer #3 (Remarks to the Author):

Zhou et al propose a methodology for improving inverse protein folding. In order to demonstrate the new method a proof of principle is performed in TnpB, a very interesting programmable nuclease with a compact size.

#general comments

1.TnpB mutant characterization would benefit of more detailed analysis:

-Offtarget activity. The authors hypothesize that a more positive surface increases activity. However, one concerning possibility is a drop in specificity. Indeed, engineering non specific charge residues has been use to refine specificity of the related cas9 protein (see Slaymaker et al, Science 2016). The authors may need to compare offtarget of these nucleases to WT or at least mention this risk of off-target increase.

-On-target assessment of programmable nuclease activity was performed with nextGen sequencing followed by indel counting. More details would need to be provided. Additionally, it would be ideal to use an established methodology to perform this analysis such as CRISPRESSO, CRISPR-A, CRISPRseek, ... due to the complexities associated with gene editing efficiency assessment.

#minor comments

"...(TnpB) is considered to be an evolutionary precursor to CRISPR-Cas system effector protein..."

- CRISPR cas has multiple origins, it seems that TnpB is a likely precursor of Cas12 CRISPR cas enzymes (Altae-Tran et al, Science 2021). I would rephrase to clarify.

1 Reply to Reviewer 1

1.1 Comment NO. 1

First, the scope of memory-efficient graph attention is much broader than the problem of inverse protein folding (IPF.) For example, AlphaFold2’s row attention includes a bias term accounting for edge-features scales quadratically and the pair-representation track scales cubically (to account for the triangular inequality). It would be interesting in future works to see how memory-efficient graph attention could be applied to tasks beyond (IPF.)

Response We would like to express our sincere gratitude to the reviewer for bringing to our attention the potential direction for future work. The memory-efficient graph attention module presents an efficient way to model graph-structured data where global dependency is crucial within the structure. We will consider applying it to other protein-related tasks and also to the study of other biomolecules. We have included this future work direction into the revised manuscript in Conclusion Section. Additionally, in response to the Comment NO. 2 by Reviewer 2, we have presented preliminary results that demonstrate the application of our proposed memory-efficient graph attention to various tasks. The obtained results are comparable to current SOTA results, indicating the potential of the proposed model architecture.

1.2 Comment NO. 2

Despite the interesting ideas introduced in the manuscript, I had difficulty following the flow of the discussion and, more importantly, the exact meaning of ‘our model.’

Response We apologize for any unclear writing. We agree that the current wording tend to cause confusion and misunderstanding. Therefore, we have renamed the proposed model as ”ProRefiner” in the revised manuscript. We hope that this clarification will make it easier to follow our work.

In our manuscript, the term ”Our model”, or the ProRefiner in the revise version, refers to the proposed sequence refining model. The model is tasked with BERT [1]-like sequence inpainting conditioned on protein structures. Specifically, we mask random residues on sequences during training. The model takes the masked partial sequences and backbone structures as input and learns to reconstruct the whole sequences. During inference, the input partial sequence to the model could be constructed in two ways, which leads to two use cases of ProRefiner. The first one involves using it as a plug-and-play plugin to refine sequences designed by other *base models*. We develop an entropy-base residue selection method to select high-quality predictions from existing models, forming a denoised partial sequence that can be utilized as input for ProRefiner. This use case is illustrated in Entire Sequence Design Section. We validate ProRefiner’s ability to consistently refine and improve the sequences designed by base models, and show its generalizability by experimenting with various

base models. The second use case is to inpaint partial sequences that are provided as fixed design context. In this setting, the fixed residues can naturally serve as an oracle partial input sequence to ProRefiner. We illustrate this use case in Partial Sequence Design Section, where models predict residues on design shells while amino acids on other positions are fixed.

In summary, the two use cases of ProRefiner differ in the starting sequences of the design, which can either be from base models or given as context. Together they demonstrate ProRefiner’s ability to leverage residue environment and generate high-quality sequences.

We have modified the model introduction part accordingly in Introduction Section in the manuscript to make the description more clear.

1.3 Comment NO. 3

For example, the ‘memory-efficient global graph attention mechanism’ discussion in the main text, starting on page 4, is not particularly informative. Either it targets readers without prior knowledge about vanilla attention, and I do not think they can follow much of the arguments, or those familiar with the attention mechanism, the discussion is relatively shallow.

Response Thank you for your valuable feedback on the discussion of ‘memory-efficient global graph attention mechanism’ part. We intended to provide background information on attention in graph domain for readers without much prior knowledge, and also explain our motivation to propose the ‘memory-efficient global graph attention’. We agree that this part of discussion is not clear enough for readers to follow. We have revised this paragraph in the manuscript for clarity and depth to the following:

Attention mechanism has been proven effective in modeling global dependencies for sequential data [2]. However, adapting attention to the graph domain is challenging. Specifically, attention mechanism calculates attention weights between any two nodes based on their features. For graphs, this requires storing and manipulating a square matrix of size equal to the number of nodes, which neglects the sparsity of graph structures and increases the memory complexity to quadratic in terms of node count, posing scalability issues [3, 4]. Some methods circumvent this by confining attention within node neighborhoods, losing the global view that makes attention powerful [5–7].

Moreover, these methods do not fully utilize edge features, as they only contribute to attention computation without the ability to be updated or influence node feature updates [3, 5, 7, 8]. However, edge features have been proven to be critical in protein structure modeling [9]. In summary, to address these limitations, we aim to design an attention-based model tailored for graphs that (1) is memory efficient, (2) maintains a global view of dependencies, and (3) fully incorporates edge features.

1.4 Comment NO. 4

Returning to the ‘model’ in the Results section, I do not understand what is reported in Table 1. On top of reporting the performance of existing models (GVP, ProteinMPNN, and ESM), I would expect one row reporting the model results, which combines the entropy-based approach and the memory-efficient attention graph attention (with and without partial sequences). I do not understand the meaning of ‘Ours GVP-GNN,’ and the same for ProteinMPNN and ESM. Moreover, what is the meaning of ‘Ours’ in Table 2? For example, is ‘Ours ESM’ trained using the same regimen as the original ESM, using 12M additional structures predicted from AlphaFold2?

Response We sincerely apologize for the confusion and misunderstanding caused. As we replied to Comment NO. 2, we have renamed the model to ProRefiner, and updated the tables accordingly as follows. We also corrected the name of the ESM model to ESM-IF1 to match the original authors’ naming convention.

Table 1 Median sequence recovery rates and nssr scores of ProRefiner with different base models on three benchmarks. ProRefiner is able to obtain good performance with relatively poor base models and achieve the highest recovery and nssr with better ones.

	CATH		TS50		Latest PDB	
	Recovery %	nssr %	Recovery %	nssr %	Recovery %	nssr %
GVP-GNN	41.27	60.81	44.02	63.59	48.02	66.23
ProRefiner + GVP-GNN	49.89	67.93	53.75	69.33	57.77	74.18
Improvement	20.89%	11.71%	22.10%	9.03%	20.30%	12.00%
ProteinMPNN	42.22	60.56	43.88	61.44	49.62	66.45
ProRefiner + ProteinMPNN	51.14	69.05	53.66	71.22	59.30	75.26
Improvement	21.13%	14.02%	22.29%	15.92%	19.51%	13.26%
ProteinMPNN-C	44.94	63.79	49.05	67.87	55.34	71.52
ProRefiner + ProteinMPNN-C	50.82	69.06	54.46	71.43	60.42	75.88
Improvement	13.08%	8.26%	11.03%	5.25%	9.18%	6.10%
ESM-IF1	55.25	71.56	55.78	72.02	63.20	77.33
ProRefiner + ESM-IF1	57.84	74.11	57.81	75.25	65.69	79.66
Improvement	4.69%	3.56%	3.64%	4.48%	3.94%	3.01%

Table 2 Median sequence recovery rates and nssr scores on EnzBench and BR_EnzBench.

	EnzBench		BR_EnzBench	
	Recovery %	nssr %	Recovery %	nssr %
GVP-GNN	41.38	57.89	29.41	47.83
ProteinMPNN-C	52.00	70.00	40.91	60.00
ProRefiner	57.89	73.68	43.48	60.87

Clarification for Table 1 The proposed ProRefiner is tasked with reconstructing the whole sequence from partial input conditioned on backbone

structure. In the first case, it can start from sequences designed by base models, serving as an add-on module to refine sequences. We demonstrate this use case in Entire Sequence Design Section of the manuscript. Thus, results in Table 1 are for entire sequence design. It is worth noting that although ProRefiner always accepts partial sequences as input, what we propose here is a sequence design pipeline with our model as a refining step and the input to this pipeline does not contain any sequence information. Therefore, we still consider this setting as entire sequence design.

Specifically, GVP-GNN[10], ProteinMPNN [9], ProteinMPNN-C and ESM-IF1 [11] are base models. "Ours [base model]", or "ProRefiner + [base model]" in the current version, denotes the sequence design pipeline that incorporates two models: the base model, which provides the initial sequence, and ProRefiner, which enhances it. The purpose of entropy-based selection within this pipeline is to identify valuable predictions from the initial sequence. We use entropy to approximate the base model's confidence in its predictions and select residues with highly confident (i.e. low entropy) predictions. These selected residues constitute a denoised, high-quality partial sequence, which is then utilized as input for ProRefiner to generate the refined sequences. Therefore, "Ours [base model]" reports the results after we employ the ProRefiner to refine the sequences generated by the corresponding base models.

Clarification for Table 2 ProRefiner can also directly generate sequences from given partial sequence environment. This scenario is discussed in Partial Sequence Design Section. Therefore, results in Table 2 are for partial sequence design and the ProRefiner row in this table reports the design results when ProRefiner is used independently to fill in the designable residues.

The training dataset for ProRefiner is CATH 4.2 training split. The details on the training of base models are provided as follows:

- **GVP-GNN** is trained on CATH 4.2 training split, the same training set as ProRefiner. We use the official code and default training parameters provided by [10] to train the model.
- **ProteinMPNN** is trained on selected PDB structures clustered into 25,361 clusters. We use the 48 edges, 0.20Å noise version of pretrained model weights released by [9].
- **ProteinMPNN-C** has the same architecture as ProteinMPNN and we train this model on the same dataset as ProRefiner.
- **ESM-IF1** is trained on CATH 4.3 training set with 16,153 structures and 12 million additional structures predicted by Alphafold2 [12]. We use the pretrained model weights released by [11].

We apologize again for any confusion caused. We have included a concise version of the above clarification in the Introduction and Results sections in the manuscript and hope it will become easier to follow.

1.5 Comment NO. 5

On top of recovery and nssr, it would be interesting to report the perplexity. If I understand correctly what the authors mean by noise, this could be related to the perplexity score.

Response Thank you a lot for your valuable feedback and your suggestion regarding reporting perplexity. While we agree that perplexity is a useful and commonly reported metric for evaluating language models, we initially chose not to report it because perplexity is defined for and applies specifically to classical autoregressive language models. Since autoregressive models generate one token based on all previous tokens, the probability of a sequence could be computed by the chain rule, and the perplexity could be calculated from the probability of the dataset. Previous inverse folding models, including the base models used in our work, generally fall into this category. However, our ProRefiner belongs to the class of masked language models, such as BERT, which are not designed to generate text in the traditional left-to-right manner, and as such, the chain rule does not apply [1, 13], making perplexity ill-defined for these models [14, 15].

Despite this, we understand the importance of providing comprehensive results and hence compute the pseudo-perplexity developed for masked language models following [15], which is not theoretically well justified but can still approximate sequence probabilities. In Table 3, ProteinMPNN and ESM-IF1 in the first group are trained on their customized dataset and other three models including our ProRefiner are trained on CATH training set. Although ProRefiner may have higher perplexity than ESM-IF1 on certain datasets, it outperforms other models by a significant margin and exhibits similar perplexity to ESM-IF1, which is trained on a much larger dataset than ours.

Table 3 Perplexity of models. Perplexity of ProRefiner is pseudo-perplexity computed following [15].

Model	CATH	TS50	Latest PDB	EnzBench	BR_EnzBench
ProteinMPNN	5.41	5.12	4.39	4.50	4.92
ESM-IF1	3.99	3.43	2.82	2.95	3.39
GVP-GNN	5.44	4.94	4.43	4.43	5.13
ProteinMPNN-C	5.21	4.52	3.83	3.89	4.47
ProRefiner	3.90	3.62	3.10	3.15	3.67

Regarding the meaning of noise, we apply the residue selection technique during inference stage to filter out less confident, noisy predictions from base models and then use ProRefiner to improve the denoised sequences. However, we understand that when computing perplexity, the input to models should be ground truth sequences without any noise, and the goal is to estimate the probability of ground truth sequences with model output. Therefore, the

effect of the proposed denoising technique may not be reflected in the perplexity metric. We sincerely appreciate your feedback and hope this explanation addresses your concerns. We have included the results of model perplexity in the Supplementary Information.

1.6 Comment NO. 6

I am unsure of the difference between the results in Table 1 and Fig3.A. I understand that Table.1 reports results about partial sequences and Fig 3.A about whole sequences. The main results reported in Table 1 should be about the entire sequence case (and not partial sequence), as this is the standard in the literature.

Response We apologize for any confusion caused by the lack of clarity in our writing. The results in Table 1 is about entire sequence design as we clarified in previous response. The results in Figure 3.A is about the ablation study on the entropy-based residue selection operation. This operation is intended to remove noisy predictions in sequences generated by base models. We would like to validate its effectiveness by removing the entropy-based selection in the entire sequence design, which means the ProRefiner is presented with the complete sequences predicted by base models. The ablation results are reported in Figure 3.A. As shown in the figure, the recovery rate without residue selection is significantly lower, indicating that ProRefiner can indeed benefit from a denoised partial sequence. More details are added to the description for Figure 3.A in Results section.

1.7 Comment NO. 7

The ablation study assessing the relative importance of global attention and partial input is run by considering on EnzBench and BR_EnzBench. Running such an ablation on CATH would have been more interesting, as it allows direct comparison with existing models.

Response

Table 4 Recovery of ProRefiner (the last row) and two ablated models, either without global attention view or without partial sequence input. The base model to produce the results on CATH is ESM-IF1. The p-values when comparing two ablated models to ProRefiner are reported.

Model		CATH		EnzBench		BR_EnzBench	
Global Attention	Partial Input	Recovery %	p-value	Recovery %	p-value	Recovery %	p-value
	✓	57.26	1.6e-5	55.56	0.0397	42.86	0.0259
✓		57.21	0.0023	56.52	0.0137	35.71	2.1e-16
✓	✓	57.84	N/A	57.89	N/A	43.48	N/A

Thank you for suggesting conducting ablation study on CATH dataset. In Figure 6.D, we report the performance of ablated models on CATH. We plot their recovery rate at different masking percentages to compare their robustness to input sequence noise. Additionally, we have added the ablation results on CATH with base model ESM-IF1 to Table 3 of the manuscript. We conduct paired two-sided t-test to compare the ablated models with ProRefiner. It is observed that we get p-value < 0.05 on all datasets, indicating a significant improvement of introducing the global attention view and input partial sequence. We have updated the Table 3 of the manuscript accordingly.

1.8 Comment NO. 8

The global graph attention model updates the node and edge features. I have a comment regarding the edge update on Equation 11. A naïve update like in Eq.11 may violate the triangular inequalities, i.e., edge features that have to fulfill the triangular inequality are not independent. Such a condition is enforced in AlphaFold2 by considering a cubic update; roughly, an edge e_{ij} is updated conditioned on all k : e_{ik} e_{kj} . I am curious if the authors thought of such constraints.

Response We sincerely appreciate the reviewer for your comment on triangle inequalities. We agree that satisfying the triangle inequality constraint could be important to ensure edge features are representable as valid 3D structures. We have tried employing the triangle multiplicative update and triangle self-attention proposed by Alphafold2 [12] in our model. We replace edge update in our model with triangle multiplicative update and triangle self-attention and denote this model as ProRefiner + TriAtten. We train the models on sequences cropped to a maximum length of 128 for a proof of concept, due to the memory issue introduced by triangular update. According to Figure 1, triangular update slows down convergence on training set, and leads to overfit problem as shown by validation loss. We then remove the proposed pseudo edge feature technique and allow triangular update to operate on real edge features. This model is denoted as ProRefiner + TriAtten - PsFeat. Removing pseudo edge features further increases model complexity and results in slightly more overfitting. Inference results of the three models are presented in Table 5.

Fig. 1 Training and validation loss curve.

Table 5 Results on CATH and EnzBench. ESM-IF1 is used as the base model for inference on CATH.

	CATH		EnzBench	
	Recovery %	nssr %	Recovery %	nssr %
ProRefiner + TriAtten - PsFeat	56.05	72.80	47.37	63.64
ProRefiner + TriAtten	56.03	72.59	48.15	62.50
ProRefiner	57.37	73.73	50.00	69.57

Due to the above empirical results, we chose to employ the simple edge update manner in our current implementation. However, we believe that the triangle inequality constraint is still very important in the context of 3D structure modeling. The above overfitting problem could be addressed by incorporating more training data, adjusting training parameters or exploring more simplified model architecture to ensure the constraint [16, 17]. We leave these directions for future work. We have added a concise version of the above discussion as a remark on Eq.11 in the manuscript.

2 Reply to Reviewer 2

2.1 Comment NO. 1

The authors evaluated model performance based on sequence recovery rate and native sequence similarity recovery. However, it is unclear how well these designed sequences fold into the target structures. It is possible that a model with lower sequence recovery rate generates sequence designs better matching with input target structures. Thus, further in-silico structural validations are essential for demonstrating the effectiveness of the proposed refinement method.

Response We sincerely appreciate your suggestions on evaluating the structure recovery of target structures. We conduct structural evaluation on TS50 and CATH datasets for entire sequence design and EnzBench dataset for partial sequence design. For partial sequence design, we additionally report the performance of ESM-IF1 (previously named as ESM) and ProRefiner + ESM-IF1 (previously denoted as Ours ESM) as required in Comment NO. 5. We predict the structures of designed sequences and report their TM-score and RMSD compared to target structures, and the pLDDT computed by the folding algorithm. Due to limited resources, we utilized Alphafold2, a computationally intensive and resource-demanding method, to fold the two relatively small datasets, TS50 and EnzBench. Meanwhile, we employ ESMFold [18] for folding the largest dataset, CATH. Results are reported in Table 6 ~ Table 8.

We observe that the sequence recovery and structural recovery of the models are not positively correlated. While some methods, such as ESM-IF1 and our model (now named as ProRefiner), achieves significantly higher sequence recovery than other methods, their improvement in structure recovery is limited. Meanwhile, the discrepancies in structure recovery among the models

are significantly smaller than those in sequence recovery. Models exhibit similar levels of structure recovery, despite the variations in their performance of sequence prediction accuracy. We believe that this may be due to the fact that the recovery of target structures is a more complex metric than sequence recovery. It is influenced by a variety of factors beyond the accuracy of the predicted sequence, including the accuracy of the folding algorithm itself. We also speculate that prior models may have hit an accuracy ceiling for structure recovery when relying solely on optimizing sequence recovery as training objective, as evidenced by their similar structure recovery performance. Overcoming this may require directly optimizing structure recovery in an end-to-end framework. We plan to investigate the problem and explore this direction in future research.

Table 6 The median TM-score, RMSD and pLDDT for predicted structures on dataset TS50. Alphafold2 is used to predict the structures of sequences designed by models.

Model	TM-score \uparrow	RMSD \downarrow	pLDDT \uparrow
ProteinMPNN	0.971	0.948	95.163
ProRefiner + ProteinMPNN	0.967	0.940	94.679
ProteinMPNN-C	0.969	0.997	94.716
ProRefiner + ProteinMPNN-C	0.971	0.886	95.362
ESM-IF1	0.974	0.838	95.186
ProRefiner + ESM-IF1	0.972	0.873	94.853

Table 7 The median TM-score, RMSD and pLDDT for predicted structures on dataset CATH. ESMFold is used to predict the structures of sequences designed by models.

Model	TM-score \uparrow	RMSD \downarrow	pLDDT \uparrow
ProteinMPNN	0.861	10.851	86.501
ProRefiner + ProteinMPNN	0.848	11.151	86.223
ProteinMPNN-C	0.845	11.142	84.991
ProRefiner + ProteinMPNN-C	0.848	11.026	85.352
ESM-IF1	0.854	10.827	85.559
ProRefiner + ESM-IF1	0.852	10.897	85.665

Table 8 The median TM-score, RMSD and pLDDT for predicted structures on dataset EnzBench. Alphafold2 is used to predict the structures of sequences designed by models.

Model	TM-score \uparrow	RMSD \downarrow	pLDDT \uparrow
GVP-GNN	0.977	0.906	95.365
ProteinMPNN-C	0.983	0.812	95.598
ProRefiner	0.982	0.848	95.974
ESM-IF1	0.981	0.833	96.118
ProRefiner + ESM-IF1	0.983	0.804	96.112

While our model’s improvement in structure recovery may be limited, we believe that its high sequence recovery is still a significant achievement. Our method represents important progress on the sequence recovery front, setting the stage for future work to translate these gains into improved structure accuracy and tightly coupled sequence and structure recovery. We appreciate your insightful comments again and the aforementioned results have been incorporated into the Supplementary Information due to limited space.

2.2 Comment NO. 2

Memory-efficient global attention layer utilizes a shared pseudo edge feature for non-existing edges in order to save memory. While the authors illustrate the model’s memory advantage in Figure 6E, it is crucial to compare the predictive performance between the memory-efficient global attention layer and the original global attention layer (and preferably across several evaluation tasks). Given provided information in the manuscript, it is inconclusive on the effectiveness and usefulness of the proposed memory-efficient global attention layer.

Response We appreciate your comments on the comparison with the original global attention layer. We agree that comparing the predictive performance is important and necessary. In addition to the original Inverse Protein Folding task, we conduct experiments on the following evaluation tasks. In all tasks, we use ProRefiner to denote the models with the proposed memory-efficient global attention layer, and ProRefiner - PsFeat to denote the models using the original global attention layer, without learning pseudo edge features to approximate global attention.

- **Relative Solvent Accessibility (RSA).** We train models to predict the relative solvent accessibility of residues. Both models have 4 attention layers and model inputs are protein sequences and backbone structures. Input residue graphs are constructed as in Inverse Protein Folding. A scalar between 0 and 1 is output for each residue through a Sigmoid layer. We employ DSSP program [19] to calculate the RSA of each residue in CATH dataset, and evaluate on its testing split. Evaluation metric is the Pearson correlation coefficient between predicted and actual RSA.
- **Ligand Binding Affinity (LBA)** We predict the binding affinity of ligands to their corresponding proteins based on the co-crystallized structure of the protein-ligand complex. Both models have 4 attention layers. Model inputs are the atoms of the pocket and ligand. An atom graph is constructed for each protein-ligand pair by connecting each atom with its 64 nearest neighbors. The models are trained to predict $-\log(K)$, where K is the binding affinity in Molar units. We employ the LBA dataset from [20] split by 30% sequence identity. Evaluation metric is the Pearson correlation coefficient between predicted and actual affinity.

Fig. 2 Recovery rate and nssr scores of ProRefiner and ProRefiner - PsFeat on different benchmarks. ESM-IF1 is used as base model for inference on CATH.

- Small Molecule Properties (SMP)** We predict two properties of small molecules: dipole moment (μ) and zero point vibrational energy (ZPVE). Both models have 4 attention layers. Model inputs are the atoms of molecules. Two atoms are connected in a graph if their distance is less than 4.5 Å. The models are trained to predict the corresponding property of the molecule. We employ the SMP dataset from [20] and use mean absolute error as metric.

Table 9 Performance of ProRefiner and ProRefiner - PsFeat on 4 tasks. The metric for RSA and LBA is Pearson correlation (higher the better) and the metric for SMP tasks is mean absolute error (lower the better).

	RSA	LBA	SMP - μ	SMP - ZPVE
ProRefiner - PsFeat	0.895	0.542	0.153	4.626e-4
ProRefiner	0.889	0.539	0.166	6.112e-4

Results for Inverse Protein Folding are presented in Figure 2. We experiment on CATH dataset with base model ESM-IF1 for entire sequence design and EnzBench and BR_EnzBench datasets for partial sequence design. The results indicate that while ProRefiner shows slightly greater performance variance on certain benchmarks, its overall performance remains similar and comparable to that of ProRefiner - PsFeat. This is further validated by its comparable performance on other tasks according to results in Table 9. We have included the above results on Inverse Protein Folding task in the manuscript and the results on other tasks in the Supplementary Information.

2.3 Comment NO. 3

Details on how base models are trained and evaluated for both entire sequence design and partial sequence design are needed to better understand the performance of the proposed refinement model. For example, ProteinMPNN, to some degree, exhibits a tradeoff between sequence recovery ratio and in-silico success rate, controlled by the training noise level. Thus, it is important to provide the backbone noise ratio for the ProteinMPNN model used.

Response Thank you for your valuable suggestion. Here are the details for the base models.

- **GVP-GNN** is trained on CATH 4.2 training split, the same training set as ProRefiner. We use the official code and default parameters provided by [10] to train and evaluate the model.
- **ProteinMPNN** is trained on selected PDB structures clustered into 25,361 clusters. We use the 48 edges, 0.20Å noise version of pretrained model weights, as it is the default model option in evaluation parameters [9].
- **ProteinMPNN-C** has the same architecture as ProteinMPNN and we train this model on CATH 4.2 training split for fair comparison with our model in the partial sequence design setting.
- **ESM-IF1** is trained on CATH 4.3 training set with 16,153 structures and 12 million additional structures predicted by Alphafold2 [12]. We use the pretrained model weights released by [11].

The same base models are used for entire sequence design and partial sequence design. We have added the above information to the Results Section in the manuscript.

2.4 Comment NO. 4

For full sequence inverse folding, the model relies on base models to generate partial sequence. What is the strength of the entropy-based mask? What is the percentage of residues that are provided as context for the later model?

Response Thank you for your comments on the strength and usage of the entropy-based mask. The entropy-based mask is designed to filter out noisy and less valuable residue identity predictions made by base models. This allows ProRefiner to decode sequences conditioned on a more reliable and denoised residue environment, without being affected by errors from previous predictions. This helps to avoid the error accumulation problem that can occur with autoregressive methods. We have demonstrated the effectiveness of the entropy-based mask in Figure 3.A of the manuscript. The results show that the recovery rate is significantly lower without the entropy mask, especially when the base model is less accurate. These results indicate that the noise in previous predictions could greatly limit the quality of subsequent generation. Using entropy mask can effectively remove the noise and provide the later model with a higher quality residue environment as a starting point.

To account for the varying ability of different base models to recover native sequences, we provide different percentages of residues to ProRefiner depending on the base model being used. The percentage of residues chosen for each base model is determined based on the recovery rate on the validation set of CATH. We experimented with percentages ranging from 5% to 50% for each base model, and selected the percentage resulting in the highest recovery rate on the validation set. Specifically, we chose 10% for GVP-GNN, 10% for ProteinMPNN, 15% for ProteinMPNN-C, and 35% for ESM-IF1. These percentages are roughly correlated with the sequence recovery performance of each base model. The above details on the usage of entropy-based mask

have been included in Entire Sequence Design with Base Model Section in the manuscript.

2.5 Comment NO. 5

For partial sequence design, is ProteinMPNN retrained with partial sequence provided as input? If not, how are GVP-GNN and ProteinMPNN used for partial sequence design? Moreover, how does ESM perform on partial sequence design?

Response We apologize for the lack of clarity. We didn't retrain ProteinMPNN and use the released pretrained model instead as we replied to Comment NO. 3. We agree that many existing methods, including GVP-GNN, ProteinMPNN and ESM-IF1, are not specifically trained with partial sequence input. They are autoregressive models trained to predict the next residue in a sequence given the preceding residues as context. However, they can still be used for partial sequence design and ProteinMPNN and ESM-IF1 have built-in implementations for partial sequence generation in their released codebase. The implementation is similar to the standard autoregressive generation setup, where the model generates one residue at a time by conditioning on the previously generated residues, except that in partial sequence decoding, some of the residues are already known and fixed. Specifically, when a residue is generated, we check if it is one of the fixed residues provided in the partial sequence and if it is, we replace it with the corresponding fixed residue. Otherwise, we keep the generated one. Then, the model moves forward to generate the next residue. Different from autoregressive models, the ProRefiner is trained in a masked language model manner, which is tasked to fill in masked residues based on partial input sequences. It decodes unknown residues in parallel, without having to generate and check each residue individually. The above details can be found in Partial Sequence Design Section.

Table 10 Median sequence recovery rates and nssr scores on EnzBench and BR_EnzBench. The three models in the first group are trained on CATH training split, while ESM-IF1 is trained on a larger customized dataset.

	EnzBench		BR_EnzBench	
	Recovery %	nssr %	Recovery %	nssr %
GVP-GNN	41.38	57.89	29.41	47.83
ProteinMPNN-C	50.00	70.00	40.91	60.00
ProRefiner	57.89	73.68	43.48	60.87
ESM-IF1	60.71	77.27	52.63	71.71
ProRefiner + ESM-IF1	65.38	81.82	54.55	72.00

We report the performance of ESM-IF1 on partial sequence design in Table 10. Note that the three models in the first group in the table are trained on CATH training split and ESM-IF1 is trained on a much larger dataset. We

further apply the proposed entropy-based mask to the results of ESM-IF1 and subsequently conduct sequence refinement with ProRefiner, and denote this result as ProRefiner + ESM-IF1. The results demonstrate that while ESM-IF1 achieves higher recovery and nssr on its own, applying the proposed refinement by ProRefiner can further enhance performance, as evidenced by the improved metrics for ProRefiner + ESM-IF1. This demonstrates the broad applicability of our proposed sequence refining approach when applied to existing state-of-the-art models like ESM-IF1. The results mentioned above have been included in Supplementary Information.

2.6 Comment NO. 6

At training time, what percentage of the residues are masked as unknown?

Response We apologize for not providing enough details in the manuscript. At training time, we randomly mask 70% of the residues as unknown. Among the remaining 30% of the residues whose identity is available to the model, we randomly select 3% and replace their identity with random amino acid types. The intuition of introducing noise to the partial input sequence is that during inference, the residue environment may still contain a small amount of noise even after applying the entropy-based mask. By training with noise, we aim to teach the model not to fully rely on the input partial environment. The above discussion has been incorporated to Graph Representation of Proteins Section.

2.7 Comment NO. 7

How much overlapping is there among the three evaluation datasets?

Response Thank you for your comment regarding the dataset overlap. We have identified four structures that are shared between the CATH test split and the TS50 dataset. There are no structures that overlap between the Latest PDB dataset and the other two datasets. The above details have been included in Entire Sequence Design Section in the manuscript.

3 Reply to Reviewer 3

3.1 Comment NO. 1

Offtarget activity. The authors hypothesize that a more positive surface increases activity. However, one concerning possibility is a drop in specificity. Indeed, engineering non specific charge residues has been use to refine specificity of the related cas9 protein (see Slaymaker et al, Science 2016). The authors may need to compare offtarget of these nucleases to WT or at least mention this risk of off-target increase.

Response

Fig. 3 Indel formation at the on-target and off-target sites observed for TnpB WT and TnpB K84R. Off-target sites are chosen following [21].

We appreciate your valuable suggestion on reporting off-target activity. We conduct experiments on the TnpB variant with the highest activity, TnpB K84R, and compare its off-target effects with those of the wild-type (WT) TnpB. The results are presented in Figure 3. As shown in the figure, the mutation of the site to arginine enhances the protein’s affinity for DNA and increases its activity. This also leads to a degree of non-specific cleavage, which may compromise the nuclease’s specificity. We have included the above discussion into Application on Transposon-associated transposase B Section of the manuscript.

3.2 Comment NO. 2

On-target assessment of programmable nuclease activity was performed with nextGen sequencing followed by indel counting. More details would need to be provided. Additionally, it would be ideal to use an established methodology to perform this analysis such as CRISPRESSO, CRISPR-A, CRISPRseek, ... due to the complexities associated with gene editing efficiency assessment.

Response Thank you for your insightful comments regarding the assessment of our programmable nuclease activity. We used CRISPResso2 to analyse Indels. The parameters are as follows: minimum of 80% homology for alignment to the amplicon sequence, quantification window of 20 bp and ignoring substitutions to avoid false positives. The above details can be found in Application on Transposon-associated transposase B Section of the manuscript. We hope that these details address your concerns, and we are grateful for your valuable feedback.

3.3 Comment NO. 3

“....(TnpB) is considered to be an evolutionary precursor to CRISPR-Cas system effector protein...” - CRISPR cas has multiple origins, it seems that TnpB is a likely precursor of Cas12 CRISPR cas enzymes (Altae-Tran et al, Science 2021). I would rephrase to clarify.

Response We apologize for the lack of clarity. We have rephrased the sentence to the following in the manuscript:

Transposon-associated transposase B (TnpB) is thought to be the ancestor of Cas12, the type V CRISPR effector [22, 23].

References

- [1] Devlin J, Chang MW, Lee K, Toutanova K. Bert: Pre-training of deep bidirectional transformers for language understanding. arXiv preprint arXiv:1810.04805. 2018;.
- [2] Vaswani A, Shazeer N, Parmar N, Uszkoreit J, Jones L, Gomez AN, et al. Attention is all you need. In: Advances in neural information processing systems; 2017. p. 5998–6008.
- [3] Hussain MS, Zaki MJ, Subramanian D. Edge-augmented Graph Transformers: Global Self-attention is Enough for Graphs. CoRR. 2021;abs/2108.03348.
- [4] Bergen L, O’Donnell TJ, Bahdanau D. Systematic Generalization with Edge Transformers. In: Ranzato M, Beygelzimer A, Dauphin YN, Liang P, Vaughan JW, editors. Advances in Neural Information Processing Systems 34: Annual Conference on Neural Information Processing Systems 2021, NeurIPS 2021, December 6-14, 2021, virtual; 2021. p. 1390–1402.
- [5] Dwivedi VP, Bresson X. A Generalization of Transformer Networks to Graphs. CoRR. 2020;abs/2012.09699.
- [6] Hu Z, Dong Y, Wang K, Sun Y. Heterogeneous Graph Transformer. In: Huang Y, King I, Liu T, van Steen M, editors. WWW ’20: The Web Conference 2020, Taipei, Taiwan, April 20-24, 2020. ACM / IW3C2; 2020. p. 2704–2710.
- [7] Ingraham J, Garg V, Barzilay R, Jaakkola T. Generative models for graph-based protein design. Advances in neural information processing systems. 2019;32.
- [8] Ying C, Cai T, Luo S, Zheng S, Ke G, He D, et al. Do Transformers Really Perform Bad for Graph Representation? CoRR. 2021;abs/2106.05234.
- [9] Dauparas J, Anishchenko I, Bennett N, Bai H, Ragotte RJ, Milles LF, et al. Robust deep learning based protein sequence design using ProteinMPNN. bioRxiv. 2022;.
- [10] Jing B, Eismann S, Suriana P, Townshend RJ, Dror R. Learning from protein structure with geometric vector perceptrons. arXiv preprint arXiv:2009.01411. 2020;.

- [11] Hsu C, Verkuil R, Liu J, Lin Z, Hie B, Sercu T, et al. Learning inverse folding from millions of predicted structures. *bioRxiv*. 2022;.
- [12] Jumper J, Evans R, Pritzel A, Green T, Figurnov M, Ronneberger O, et al. Highly accurate protein structure prediction with AlphaFold. *Nature*. 2021;596(7873):583–589.
- [13] Rezaee M, Darvish K, Kebe GY, Ferraro F. Discriminative and generative transformer-based models for situation entity classification. *arXiv preprint arXiv:210907434*. 2021;.
- [14] Wang A, Cho K. BERT has a mouth, and it must speak: BERT as a Markov random field language model. *arXiv preprint arXiv:190204094*. 2019;.
- [15] Salazar J, Liang D, Nguyen TQ, Kirchhoff K. Masked language model scoring. *arXiv preprint arXiv:191014659*. 2019;.
- [16] Pitis S, Chan H, Jamali K, Ba J. An inductive bias for distances: Neural nets that respect the triangle inequality. *arXiv preprint arXiv:200205825*. 2020;.
- [17] Hsieh CK, Yang L, Cui Y, Lin TY, Belongie S, Estrin D. Collaborative metric learning. In: *Proceedings of the 26th international conference on world wide web*; 2017. p. 193–201.
- [18] Lin Z, Akin H, Rao R, Hie B, Zhu Z, Lu W, et al. Language models of protein sequences at the scale of evolution enable accurate structure prediction. *bioRxiv*. 2022;.
- [19] Kabsch W, Sander C. Dictionary of protein secondary structure: pattern recognition of hydrogen-bonded and geometrical features. *Biopolymers: Original Research on Biomolecules*. 1983;22(12):2577–2637.
- [20] Townshend RJ, Vögele M, Suriana P, Derry A, Powers A, Laloudakis Y, et al. Atom3d: Tasks on molecules in three dimensions. *arXiv preprint arXiv:201204035*. 2020;.
- [21] Nakagawa R, Hirano H, Omura SN, Nety S, Kannan S, Altae-Tran H, et al. Cryo-EM structure of the transposon-associated TnpB enzyme. *Nature*. 2023;616(7956):390–397.
- [22] Makarova KS, Wolf YI, Iranzo J, Shmakov SA, Alkhnbashi OS, Brouns SJ, et al. Evolutionary classification of CRISPR–Cas systems: a burst of class 2 and derived variants. *Nature Reviews Microbiology*. 2020;18(2):67–83.

- [23] Altae-Tran H, Kannan S, Demircioglu FE, Oshiro R, Nety SP, McKay LJ, et al. The widespread IS200/IS605 transposon family encodes diverse programmable RNA-guided endonucleases. *Science*. 2021;374(6563):57–65.

Reviewer #1 (Remarks to the Author):

I want to thank the authors for their efforts to answer my questions, and I don't have any further concerns.

Reviewer #2 (Remarks to the Author):

Thank you for addressing my concerns in my previous comment and validating the pipeline with additional experiments. Overall, this paper introduces a refinement model that further improves inverse folding performance. In addition, the memory-efficient global attention mechanism provides a possible way to reduce the memory usage, without significantly trading off on the performance of global attention. There are two minor points that would require further clarifications:

1. How is AlphaFold2 used for structural evaluation? Does it utilize only MSA of the designed sequence, or does it utilize both MSA and searched templates?
2. Comparing the results in Supplementary Table 4 with the results in Supplementary Table 3 (or Supplementary Table 5), there is a significant drop in structural recovery performance in terms of both TM score and RMSD. What are the main reasons behind this drop in performance? Also, it would be useful to confirm if this is the effect of switching in datasets or the effect of switching in structure prediction models.

Reviewer #3 (Remarks to the Author):

Authors revisions solved all my concerns. No more comments from my side.

1 Reply to Reviewer 1

We sincerely appreciate your time and effort in reviewing our manuscript. We are grateful for your valuable comments, which have helped us refine the quality and clarity of our manuscript.

2 Reply to Reviewer 2

2.1 Comment NO. 1

How is AlphaFold2 used for structural evaluation? Does it utilize only MSA of the designed sequence, or does it utilize both MSA and searched templates?

Response Thank you a lot for your valuable feedback. We run Alphafold2 with both MSA and searched templates. We run relaxation for all 5 models and take the structures ranked first as the predicted structures. The above details, including more details on ESMFold usage, have been included in the Supplementary Information.

2.2 Comment NO. 2

Comparing the results in Supplementary Table 4 with the results in Supplementary Table 3 (or Supplementary Table 5), there is a significant drop in structural recovery performance in terms of both TM-score and RMSD. What are the main reasons behind this drop in performance? Also, it would be useful to confirm if this is the effect of switching in datasets or the effect of switching in structure prediction models.

Response We sincerely appreciate your comments on the structure recovery drop in CATH dataset. To confirm if this drop is due to using ESMFold instead of Alphafold2, we use ESMFold to evaluate the other two datasets, TS50 and EnzBench. The results are shown in Table 1 and 2. It could be observed that switching to ESMFold leads to a slight performance drop in term of TM-score and RMSD and a significant drop in pLDDT.

Table 1 The median TM-score, RMSD and pLDDT for predicted structures on dataset TS50. ESMFold is used to predict the structures of sequences designed by models.

Model	TM-score \uparrow	RMSD \downarrow	pLDDT \uparrow
ProteinMPNN	0.957	1.177	85.991
ProRefiner + ProteinMPNN	0.951	1.466	85.199
ProteinMPNN-C	0.947	1.456	83.127
ProRefiner + ProteinMPNN-C	0.949	1.462	85.209
ESM-IF1	0.957	1.344	85.529
ProRefiner + ESM-IF1	0.956	1.285	84.758

We have observed that the CATH dataset contains many structures with missing coordinates, which we handle by padding them with 0 coordinates during

Table 2 The median TM-score, RMSD and pLDDT for predicted structures on dataset EnzBench. ESMFold is used to predict the structures of sequences designed by models.

Model	TM-score \uparrow	RMSD \downarrow	pLDDT \uparrow
GVP-GNN	0.953	1.418	84.072
ProteinMPNN-C	0.954	1.302	84.809
ProRefiner	0.960	1.400	84.823
ESM-IF1	0.960	1.184	85.094
ProRefiner + ESM-IF1	0.954	1.267	85.217

Table 3 The median TM-score, RMSD and pLDDT for predicted structures on dataset CATH (all) and the subset of structures without missing coordinates (valid). ESMFold is used to predict the structures of sequences designed by models.

Model	TM-score \uparrow		RMSD \downarrow	
	all	valid	all	valid
ProteinMPNN	0.861	0.866	10.851	2.872
ProRefiner + ProteinMPNN	0.848	0.837	11.151	3.316
ProteinMPNN-C	0.845	0.833	11.142	3.510
ProRefiner + ProteinMPNN-C	0.848	0.829	11.026	3.190
ESM-IF1	0.854	0.846	10.827	3.166
ProRefiner + ESM-IF1	0.852	0.841	10.897	3.362

evaluation, while the TS50 and EnzBench datasets do not have this issue. To validate if the presence of missing portions in structures will impact the comparison of these structures with the predicted ones, we exclude structures with missing coordinates from the CATH dataset. As a result, we obtained a subset of 282 structures. The structure recovery on this subset can be found in Table 3. It can be observed that the RMSD metric, which is known to be more sensitive to local structural variations [1], is largely improved on the valid subset. On the other hand, the TM-score values remain roughly the same.

We further plot the scatter plot between sequence length and TM-score with ESMFold as the folding algorithm. To simplify the presentation, we focus on two representative models, namely ProteinMPNN and ESM-IF1, which are trained on distinctly different training sets. As illustrated in Fig. 1 a and b, models tend to obtain higher TM-scores on longer sequences, while displaying a larger performance variance on shorter ones. This is further supported by Fig. 1 c and d, where we investigate the deviation of C α coordinates of residues on different secondary structures after superimposing the predicted structures onto native ones using the Kabsch algorithm [2, 3]. Notably, residues located on helices and coils from shorter sequences tend to exhibit a greater deviation. The observed larger structural variations on helix residues align with our manuscript’s findings, where we identified lower sequence recovery rates in helical regions. The performance decline on coils might be attributed to

Fig. 1 a-b. Scatter plot between sequence length and TM-score of sequences predicted by ProteinMPNN (a) and ESM-IF1 (b) folded by ESMFold. For CATH dataset, only the valid subset without missing coordinates is plotted. **c-d.** The deviation of $\text{C}\alpha$ coordinates of residues on different secondary structures after superimposing the predicted structures by ProteinMPNN (c) and ESM-IF1 (d) onto native ones. "Short" refers to residues from sequences with a length less than 100, while "Long" refers to those longer than 100 residues.

the inherent challenge in accurately modeling coil structures by folding models, depicted by lower pLDDT values (median pLDDT of 82.64 for coils versus 90.81 and 91.92 for helices and strands). In light of this, we count the sequences in each dataset. Within the CATH valid subset, 32.6% sequences have a length less than 100 and among these short sequences 87.8% residues are located on helices or coils. TS50 and EnzBench have 16.0% and 2.0% short sequences and 75.5% and 76.1% residues are located on helices or coils respectively. Based on the above discussion, we believe that the presence of a higher proportion of short sequences containing helices and coils also contributes to the structure recovery drop on CATH dataset.

In summary, several factors might contribute to the observed performance gap: (1) ESMFold model yields significantly lower pLDDT values and slightly lower performance on TM-score and RMSD metrics; (2) the presence of structures with missing coordinates leads to increased RMSD values; (3) the dataset contains a large portion of short sequences with structures that are generally difficult for models to recover. We would like to express our gratitude for bringing our attention to this performance discrepancy. It would be a promising future direction to focus on enhancing the model's performance specifically on short sequences, which would greatly benefit the design of small proteins or peptides such as cathelicidins. It is also worth further research to explore other

potential factors contributing to this issue, which will help guide advancements in inverse protein folding models.

3 Reply to Reviewer 3

We would like to sincerely thank you for taking the time to review our work. Your feedback has been invaluable in improving the clarity and comprehensiveness of our work.

References

- [1] Zhang Y, Skolnick J. Scoring function for automated assessment of protein structure template quality. *Proteins: Structure, Function, and Bioinformatics*. 2004;57(4):702–710.
- [2] Kabsch W. A solution for the best rotation to relate two sets of vectors. *Acta Crystallographica Section A: Crystal Physics, Diffraction, Theoretical and General Crystallography*. 1976;32(5):922–923.
- [3] Kabsch W. A discussion of the solution for the best rotation to relate two sets of vectors. *Acta Crystallographica Section A: Crystal Physics, Diffraction, Theoretical and General Crystallography*. 1978;34(5):827–828.